# $O(n)$ Connections are Expressive Enough: Universal Approximability of Sparse Transformers

**Chulhee Yun**
MIT
chulheey@mit.edu

**Yin-Wen Chang**
Google Research NY
yinwen@google.com

**Srinadh Bhojanapalli**
Google Research NY
bsrinadh@google.com

**Ankit Singh Rawat**
Google Research NY
ankitsrawat@google.com

**Sashank J. Reddi**
Google Research NY
sashank@google.com

**Sanjiv Kumar**
Google Research NY
sanjivk@google.com

## Abstract

Recently, Transformer networks have redefined the state of the art in many NLP tasks. However, these models suffer from quadratic computational cost in the input sequence length $n$ to compute pairwise attention in each layer. This has prompted recent research into *sparse Transformers* that sparsify the connections in the attention layers. While empirically promising for long sequences, fundamental questions remain unanswered: Can sparse Transformers approximate any arbitrary sequence-to-sequence function, similar to their dense counterparts? How does the sparsity pattern and the sparsity level affect their performance? In this paper, we address these questions and provide a *unifying framework* that captures existing sparse attention models. We propose sufficient conditions under which we prove that a sparse attention model can *universally approximate* any sequence-to-sequence function. Surprisingly, our results show that sparse Transformers with only $O(n)$ connections per attention layer can approximate the same function class as the dense model with $n^2$ connections. Lastly, we present experiments comparing different patterns/levels of sparsity on standard NLP tasks.

## 1 Introduction

Transformer networks [28] and their variants [31] have played a key role in the recent advancement of the state of the art in many natural language processing tasks, such as machine translation [28], language modeling [23, 24], and question answering [10, 17, 31]. The key component of these networks is the self-attention layer [1, 18], which updates the embeddings of the input tokens based on their context. Naturally, the self-attention layer also plays the key role in the analysis of Transformers [3, 4, 12, 20, 33]; for example, Yun et al. [33] show that Transformers can approximate any continuous sequence-to-sequence functions (i.e., universal approximation), by proving that self-attention layers can compute *contextual mappings* of the input embeddings.

On the other hand, the self-attention layer is also the main bottleneck in scaling these models. It involves computation of *pairwise* inner products between input tokens, which results in quadratic computational complexity $O(n^2)$ in the length of the input sequence $n$. To mitigate this issue, researchers have developed methods to *sparsify* the pairwise interactions/connections in self-attention layers to reduce the computational complexity and/or improve model interpretability, and have shown successful empirical results on tasks with long sequence lengths [2, 6, 8, 9, 11, 16, 22, 25, 26, 32, 34, 35]. For example, Child et al. [6] propose sparse Transformers for sequence generation. One of the sparsity patterns considered in [6] is the STRIDED pattern, where the sparse attention layers alternate between two patterns: each token attends to only i) $w$ local neighbors, and then ii) one after

every $w$ tokens in a strided manner. By choosing $w = O(\sqrt{n})$, they propose sparse attention layers with $O(n^{3/2})$ connections and show improvements on both speed and performance over the dense Transformer.

In the existing results, the rule of thumb for designing sparsity patterns (e.g., STRIDED) is connectivity; the intuition is that if each token can attend to the other tokens in multiple "hops," then the resulting sparse Transformers do not lose much expressive power. However, there has been *no formal justification* for this intuition. How does sparsifying the interaction in the self-attention layers affect the model's expressive power and ability to learn? What are the sparsity levels at which the model still retains its rich expressive power, and how is it affected by the sparsity pattern? Such fundamental questions about sparse attention models still remain unanswered.

## 1.1 Summary of contributions

In this paper, we take the first step towards a theoretical understanding of sparse Transformers.

- We propose a unified framework to analyze sparse Transformers, which generalizes the existing approaches that sparsify attention layers (§ 3.1).
- We propose a set of intuitive conditions on the sparsity pattern (Assumption 1) and the probability map (Assumption 2). Then, in Theorem 1, we show that Sparse Transformers, of fixed width and arbitrary depth, satisfying these conditions are universal approximators of any continuous sequence-to-sequence functions for any given fixed sequence length (§ 3.2 and § 3.3).
- We next show some examples of existing sparse Transformers [2, 6, 8, 9, 11, 34, 35] that satisfy these conditions, and hence have universal approximability (§ 3.4). Surprisingly, we show that there are sparse Transformers with only $O(n)$ connections per self-attention layer (instead of $n^2$) that have enough expressive power to approximate arbitrary continuous functions (Corollary 2).
- We report experimental results on standard NLP tasks using sparse Transformers, comparing different sparsity patterns/levels (§ 5).

## 2 Preliminaries and related works

In this section, we summarize the notation we will use throughout the paper, give a brief overview of Transformers, and then discuss existing efforts to sparsify the self-attention mechanism.

### 2.1 Notation

For a positive integer $a$, we denote $[a] = \{1, 2, \ldots, a\}$. For any vector $\boldsymbol{v} \in \mathbb{R}^d$, let $v_j$ denote its $j$-th coordinate. For any matrix $\boldsymbol{A} \in \mathbb{R}^{d \times n}$, let $\boldsymbol{A}_j$ denote its $j$-th column, and $\boldsymbol{A}_{\mathcal{S}}$ denote the submatrix consisting of columns of $\boldsymbol{A}$ in an index set $\mathcal{S} \subseteq [n]$. We use $\|\boldsymbol{A}\|_p$ to denote the entry-wise $\ell^p$ norm of $\boldsymbol{A}$. Let $\sigma_{\mathrm{S}}[\cdot]$ be the softmax operator, which takes a matrix as input and applies softmax operation to each column of the matrix, which results in a column stochastic matrix.

### 2.2 Transformers and their universal approximation power

A Transformer network, consisting of multiple layers of Transformer blocks, implements a sequence-to-sequence function that maps $\mathbb{R}^{d \times n}$ to $\mathbb{R}^{d \times n}$. A Transformer Block (TB) consists of two layers: a self-attention layer and a token-wise feed-forward layer, and both layers have an identity skip connection. More concretely, for an input $\boldsymbol{X} \in \mathbb{R}^{d \times n}$ consisting of $d$-dimensional embeddings of $n$ tokens, a Transformer block consists of the following two layers:

$$\mathrm{Attn}(\boldsymbol{X}) = \boldsymbol{X} + \boldsymbol{W}_O \begin{bmatrix} \mathrm{Head}^1(\boldsymbol{X}) \\ \vdots \\ \mathrm{Head}^h(\boldsymbol{X}) \end{bmatrix}; \;\; \mathrm{Head}^i(\boldsymbol{X}) = \boldsymbol{W}_V^i \boldsymbol{X} \cdot \sigma_{\mathrm{S}}[(\boldsymbol{W}_K^i \boldsymbol{X})^T \boldsymbol{W}_Q^i \boldsymbol{X}] \quad \text{(1a)}$$

$$\mathrm{TB}(\boldsymbol{X}) = \mathrm{Attn}(\boldsymbol{X}) + \boldsymbol{W}_2 \cdot \mathrm{ReLU}(\boldsymbol{W}_1 \cdot \mathrm{Attn}(\boldsymbol{X})), \quad \text{(1b)}$$

where $\boldsymbol{W}_O \in \mathbb{R}^{d \times mh}$, $\boldsymbol{W}_V^i, \boldsymbol{W}_K^i, \boldsymbol{W}_Q^i \in \mathbb{R}^{m \times d}$, $\boldsymbol{W}_2 \in \mathbb{R}^{d \times r}$, and $\boldsymbol{W}_1 \in \mathbb{R}^{r \times d}$. Although our analysis and experiments rely on bias vectors, we omit those in (1) for simplicity.

To endow the network with information about the position of input tokens, it is common to add a positional embedding $\boldsymbol{E} \in \mathbb{R}^{d \times n}$ to the input $\boldsymbol{X}$ before feeding it to the network. The positional

embedding $\boldsymbol{E}$ can be fixed [28] or trainable [10]; we consider the latter. Using a trainable $\boldsymbol{E}$, $\mathcal{T}^{h,m,r}$ is defined to be a class of functions of the form $\boldsymbol{X} \mapsto t(\boldsymbol{X} + \boldsymbol{E})$, where $t$ is a composition of any number of Transformer blocks with $h$ attention heads of head size $m$, and hidden layers of width $r$. Thus, $\mathcal{T}^{h,m,r}$ is a class of Transformers with a fixed *width* while the depth can be arbitrary.

Further, let $\mathcal{F}$ be the class of continuous functions $f : \mathbb{D} \to \mathbb{R}^{d \times n}$ defined on any compact domain $\mathbb{D} \subset \mathbb{R}^{d \times n}$, where continuity is defined with respect to the entry-wise $\ell_p$ norm ($1 \leq p < \infty$). Yun et al. [33, Theorem 3] show that $\mathcal{T}^{2,1,4}$ can *universally approximate* $\mathcal{F}$. More precisely, for any $f \in \mathcal{F}$, $\epsilon > 0$ and $1 \leq p < \infty$, there exists a function $g \in \mathcal{T}^{2,1,4}$ such that $\mathsf{d}_p(f,g) := (\int_{\mathbb{D}} \|f(\boldsymbol{X}) - g(\boldsymbol{X})\|_p^p \, d\boldsymbol{X})^{1/p} \leq \epsilon$. Our goal in this paper is to study, in a similar manner, the expressive power of *sparse* Transformers.

### 2.3 Sparse Transformers

As seen in Eq. (1a), the self-attention layer involves computing the inner product between each pair of tokens, which we will refer to as the **attention score matrix** $\boldsymbol{A}^i := (\boldsymbol{W}_K^i \boldsymbol{X})^T \boldsymbol{W}_Q^i \boldsymbol{X} \in \mathbb{R}^{n \times n}$. This leads to quadratic computational complexity in $n$, which makes it expensive to apply Transformers to tasks with long sequence lengths. One popular approach to mitigate this problem is to *sparsify* the self-attention layers. We sub-classify sparse Transformers into three categories and summarize them below. For a more extensive summary, please see a recent survey [27].

The first category reduces computation by making $\boldsymbol{A}^i$ sparse in a *pre-determined* manner. Each token in the sequence only attends to a fixed smaller set of other tokens instead of the whole sequence [2, 6, 22]. In some papers, auxiliary tokens are added to improve connectivity between existing tokens while maintaining sparsity [11, 32]. One drawback of these approaches is that the sparsity pattern is independent of input, so it cannot adapt to the data. To remedy this issue, [26] proposes to learn local attention span from data. In a concurrent paper, Zaheer et al. [34] propose the BIGBIRD sparsity pattern which falls into this category. For BIGBIRD, the authors show its theoretical properties such as universal approximation and Turing completeness, as well as its superior empirical performance. We note that our paper focuses on universal approximation for a *broader* class of sparse Transformers, by proposing a unifying framework to analyze them.

The second category studies making $\boldsymbol{A}^i$ sparse *after* the full $\boldsymbol{A}^i$ has been computed [8, 9, 35]. Here, the focus is not on the computational gain via sparsity, because the full score matrix $\boldsymbol{A}^i$ has to be computed first; rather, the goal here is to make attention layers more interpretable, as well as to improve performance. This line of works modifies $\sigma_S$ in (1a) to other probability maps, by using top-$k$ elements or adopting sparser variants such as sparselin-gen or $\alpha$-entmax [15, 21]. Compared to the first category, this approach has an advantage that sparsity patterns are adaptive to data.

The last category attempts to get the best of both worlds. This line of works tries to learn sparsity patterns from data using extra components predicting the connection between tokens, e.g., $k$-means clustering [25], LSTM [16], or locality-sensitive hashing [14]. This way, one can adaptively determine the sparsity patterns before computing the score matrix. However, the drawback of this approach is that one needs extra computation to train/run these additional components, which may be expensive.

## 3 Universal approximation theorem for sparse Transformers

In this section, we derive a unifying framework to study sparse Transformers. We then propose a set of conditions on the sparse self-attention layers, and prove that the sparse Transformers satisfying theses conditions are universal approximators of any continuous sequence-to-sequence functions. Finally, we show some examples of existing sparse Transformers that satisfy these conditions.

### 3.1 A unifying framework for sparse Transformers

We modify the Transformer block in (1) to the following sparse Transformer block (STB):

$$\mathrm{SAttn}^l(\boldsymbol{X}) = \boldsymbol{X} + \boldsymbol{W}_O \begin{bmatrix} \mathrm{SHead}^{1,l}(\boldsymbol{X}) \\ \vdots \\ \mathrm{SHead}^{h,l}(\boldsymbol{X}) \end{bmatrix}, \quad \mathrm{SHead}^{i,l}(\boldsymbol{X})_k = \boldsymbol{W}_V^i \boldsymbol{X}_{\mathcal{A}_k^l} \cdot \rho[(\boldsymbol{W}_K^i \boldsymbol{X}_{\mathcal{A}_k^l})^T \boldsymbol{W}_Q^i \boldsymbol{X}_k]$$

$$\mathrm{STB}^l(\boldsymbol{X}) = \mathrm{SAttn}^l(\boldsymbol{X}) + \boldsymbol{W}_2 \cdot \mathrm{ReLU}(\boldsymbol{W}_1 \cdot \mathrm{SAttn}^l(\boldsymbol{X})), \tag{2}$$

where the sets $\mathcal{A}_k^l \subseteq [n]$, for $k \in [n]$ and $l \in [p]$, define the $p$ sparsity patterns (formally defined below), which are indexed by $l \in [p]$. Moreover, the parameter dimensions stay the same as in (1).

Note that there are three main modifications from the dense Transformer.

- (*Cycling blocks*) There are superscripts $l \in [p]$ added to the symbols such as SAttn. Unlike dense Transformers, some sparse Transformers **cycle** through $p$ different patterns. For example, the STRIDED pattern [6] described in § 1 alternates between two different patterns, which corresponds to $p = 2$. We add the superscript $l$ to include such cases in our formulation. We assume that the layers in a sparse Transformer cycle through $\mathrm{STB}^1, \ldots, \mathrm{STB}^p$.

- (*Sparsity patterns*) Note that $\mathrm{SHead}^{i,l}(\boldsymbol{X})_k$ denotes the $k$-th column of the $i$-th sparse attention head. Unlike dense Transformers, the inner product of the $k$-th query vector $\boldsymbol{W}_Q^i \boldsymbol{X}_k$ is taken only with $\boldsymbol{W}_K^i \boldsymbol{X}_{\mathcal{A}_k^l}$, the key vectors of tokens in the set $\mathcal{A}_k^l \subseteq [n]$. Hence, instead of all $n$ tokens, the $k$-th token computes attention scores with only tokens in $\mathcal{A}_k^l$. For $l \in [p]$, we refer to the collection of the index sets $\{\mathcal{A}_k^l\}_{k \in [n]}$, or simply $\{\mathcal{A}_k^l\}$, as a **sparsity pattern**. As a result, $\mathrm{SHead}^{i,l}(\boldsymbol{X})_k$ is a linear combination of columns in $\boldsymbol{W}_V^i \boldsymbol{X}_{\mathcal{A}_k^l}$, rather than the whole sequence.

- (*Probability map*) After computing the attention score matrix, the dense Transformer (1) uses the softmax operator $\sigma_{\mathrm{S}}$ to get a column stochastic matrix. In the sparse Transformers, we generalize $\sigma_{\mathrm{S}}$ to $\rho$. The **probability map** $\rho$ is any map that takes a matrix as input and outputs a column stochastic matrix.

As a sanity check, by choosing $p = 1$, $\mathcal{A}_k^1 = [n]$ for all $k \in [n]$, and $\rho = \sigma_{\mathrm{S}}$, we recover the dense Transformer (1). Note also that the sparse Transformer formulation covers the first and second categories of existing results discussed in § 2.3. The first category corresponds to choosing a predetermined sparsity pattern(s) $\{\mathcal{A}_k^l\}$, while setting $\rho = \sigma_{\mathrm{S}}$. The second category corresponds to opting for a probability map $\rho$ other than softmax $\sigma_{\mathrm{S}}$, while maintaining $\mathcal{A}_k^1 = [n]$ for all $k \in [n]$.

In this paper, we assume for simplicity that all sparse attention heads $\mathrm{SHead}^{1,l}, \ldots, \mathrm{SHead}^{h,l}$ in a single layer have identical sparsity patterns $\{\mathcal{A}_k^l\}$. However, since our result only requires two sparse attention heads per layer (as we will see in Theorem 1), our result can be easily extended to the case that allows multiple sparsity patterns in a single layer.

Similar to $\mathcal{T}^{h,m,r}$ in § 2.2, we define the class of functions represented by sparse Transformers. We hide the dependence of this class on the sparsity patterns and probability map to simplify the notation.

$$\mathcal{ST}^{h,m,r} := \{\boldsymbol{X} \mapsto t(\boldsymbol{X} + \boldsymbol{E}) \mid t \text{ is a composition of cycling sparse Transformer blocks } \mathrm{STB}^l,$$
$$\text{each with } h \text{ heads of head size } m \text{ and hidden layer size } r,$$
$$\text{and positional embedding } \boldsymbol{E} \in \mathbb{R}^{d \times n} \text{ is trainable}\}. \tag{3}$$

## 3.2 Conditions on sparsity patterns and probability map

In this section, we define a set of conditions on the sparsity patterns $\{\mathcal{A}_k^l\}$ and the probability map $\rho$ that ensures that the sparse Transformer universally approximate the function class $\mathcal{F}$ (cf. § 2.2).

For $k \in [n]$ and the index sets $\{\mathcal{A}_k^l\}_{l \in [p]}$, we define a sequence of sets $\{\mathcal{S}_k^t\}_{t \geq 1}$ in a recursive way:

$$\mathcal{S}_k^1 := \mathcal{A}_k^1, \ \ \mathcal{S}_k^t := \bigcup_{j \in \mathcal{A}_k^{(t-1) \bmod p + 1}} \mathcal{S}_j^{t-1}.$$

The set $\mathcal{S}_k^t$ is the set of all tokens that the $k$-th token can directly/indirectly attend to, after $t$ sparse attention layers with sparsity patterns cycling through $\{\mathcal{A}_k^1\}, \{\mathcal{A}_k^2\}, \ldots, \{\mathcal{A}_k^p\}$. We now state our conditions on sparsity patterns.

**Assumption 1.** *The sparsity patterns* $\{\mathcal{A}_k^l\}$ *satisfy the following:*

1. *For all $k \in [n]$ and $l \in [p]$, we have $k \in \mathcal{A}_k^l$.*

2. *There exists a permutation $\gamma : [n] \to [n]$ such that, for all $i \in [n-1]$, $\gamma(i) \in \bigcup_{l=1}^p \mathcal{A}_{\gamma(i+1)}^l$.*

3. *There exists a finite $s \in \mathbb{N}$ such that $s = \min\{u \mid \mathcal{S}_k^u = [n] \text{ for all } k \in [n]\}$.*

Assumption 1.1 is equivalent to saying that every token always attends to itself. Assumption 1.2 requires that there is a chain of *direct* connections that covers all $n$ tokens; note that the set $\bigcup_{l=1}^{p} \mathcal{A}_{\gamma(i+1)}^{l}$ is the set of all tokens that the $\gamma(i+1)$-th token directly attends to. To elaborate more about the chain, consider a directed graph with $n$ vertices corresponding to the $n$ tokens. For any $j \in \bigcup_{l=1}^{p} \mathcal{A}_{k}^{l}$, we add a directed edge $j \to k$. Given a graph constructed this way, Assumption 1.2 requires that the graph has a Hamiltonian path $\gamma(1) \to \gamma(2) \to \cdots \to \gamma(n)$. Assumption 1.3 requires that after $s$ sparse attention layers, every token can attend to all the other tokens, either directly or indirectly.

As we discuss in § 3.4, the statements in Assumption 1 are natural enough to be satisfied by many existing sparsity patterns studied in the literature. In fact, Assumption 1.3 is *necessary* for universal approximation. If $p = 1$, $n = 2$, $\mathcal{A}_1^1 = \{1\}$ and $\mathcal{A}_2^1 = \{1, 2\}$, then the first token never attends to the second, so this sparse Transformer cannot approximate a function whose first output token is dependent on both input tokens. The other two assumptions are required in parts of our proof, which involve "propagating information" over all the tokens in a sequential manner.

We now state the assumption on the probability map $\rho[\cdot]$. For this, we define $\sigma_{\mathrm{H}}[\cdot]$ to be the hardmax operator, which outputs the one-hot representation of the $\arg\max$ entry for each column of the input matrix. Since $\rho$ is a column-wise operator that outputs a column-stochastic matrix, we state the assumption for the operation of $\rho$ on a single column.

**Assumption 2.** *For any $\zeta > 0$ and $\eta \in (0, 1]$, $\exists\, t > 0$ such that, for any column input $v$ satisfying $v_{j^*} - \max_{j \neq j^*} v_j \geq \zeta$ (where $j^* = \arg\max_j v_j$), we have $\rho[tv]_{j^*} \geq 1 - \eta$ and $\sum_{j \neq j^*} \rho[tv]_j \leq \eta$.*

Assumption 2 requires that, for inputs that have some margin between the unique maximum entry and the other entries, $\rho[\cdot]$ can closely approximate the behavior of the hardmax operator by scaling its input by a positive factor $t$. This assumption is satisfied by softmax $\sigma_{\mathrm{S}}$ and other sparse variants such as sparselin-gen and $\alpha$-entmax, as we show in § B of the supplementary material.

It is straightforward to check that the dense Transformer, which corresponds to $p = 1$, $\mathcal{A}_k^1 = [n]$, and $\rho[\cdot] = \sigma_{\mathrm{S}}[\cdot]$ in our framework, satisfies both Assumptions 1 and 2.

### 3.3 Sparse Transformers are universal approximators

The key justifying intuition for adopting sparse attention layers is that, if each token can attend to the other tokens in multiple hops[1], then these models do not lose too much expressive power. However, turning this intuition into a rigorous analysis is not straightforward. Moreover, recent results show that *limited width* can render universal approximation *impossible* even with arbitrary depth [13, 19], highlighting the challenges in analyzing sparse (limited "width") Transformers.

We now state our main theorem, which shows that if the sparsity patterns $\{\mathcal{A}_k^l\}$ and the probability map $\rho$ satisfy Assumptions 1 and 2, sparse Transformers with $h = 2$ attention heads of size $m = 1$, and hidden layer width $r = 4$ are universal approximators of continuous sequence-to-sequence functions on any compact domain (recall that $\mathcal{F}$ denotes the class of such continuous functions).

**Theorem 1.** *Consider any $f \in \mathcal{F}$, and the class of sparse Transformers $\mathcal{ST}^{2,1,4}$ (cf. (3)) with the underlying sparse attention layers satisfying Assumptions 1 and 2. Then, for any $\epsilon > 0$ and $1 \leq p < \infty$, there exists a function $g \in \mathcal{ST}^{2,1,4}$ such that*

$$\mathsf{d}_p(f, g) := \left( \int_{\mathbb{D}} \|f(\boldsymbol{X}) - g(\boldsymbol{X})\|_p^p \, d\boldsymbol{X} \right)^{1/p} \leq \epsilon.$$

As discussed earlier, dense Transformers satisfy Assumptions 1 and 2, which means that Theorem 1 subsumes the existing result [33] for dense Transformers. We note that the required $h$, $m$, and $r$ in Theorem 1 are independent of $d$, $n$, or the sparsity patterns. We provide a high-level proof sketch of Theorem 1 in § 4.1. There, we also discuss how many layers are sufficient for $\epsilon$-approximation of $f$, and show that Theorem 1 requires only $p$ times more self-attention layers than Yun et al. [33].

We would like to emphasize that Theorem 1 provides the first *formal evidence* that well-designed sparse attention layers do not limit Transformer's universal approximation power. In § 3.4, we show a surprising fact that some existing sparse self-attention layers with only $O(n)$ connections (as opposed to $n^2$ in regular self-attention layers) retain enough expressive power to approximate $\mathcal{F}$. Combined with the number of layers analyzed in § 4.1, this means that our analysis reduces the

connections per layer from $n^2$ to $O(n)$, with only $p$ times more attention layers. This advantage of sparse Transformers over their dense counterpart becomes even stronger with increasing sequence length $n$, providing a theoretical support for the adoption of sparsity for tasks with long sequence lengths.

On a final note, Theorem 1 views the sequence length $n$ as a fixed constant. Hence, our result does not contradict a recent paper by Hahn [12] which studies the limitation of Transformers for varying $n$. Also, our analysis applies to the encoder part of the Transformer network [28].

### 3.4 Analysis of existing sparse Transformers

By Theorem 1, any sparse Transformer that satisfies our Assumptions 1 and 2 has universal approximation ability. In this section, we give some examples of such sparse Transformers.

Child et al. [6] propose two kinds of 2-step sparsity patterns (i.e., $p = 2$) for sequence generation tasks, namely STRIDED and FIXED patterns. We consider the extension of their auto-regressive patterns (i.e., attending only to past tokens) to the whole sequence. In the STRIDED pattern, a token first attends to its $w$ neighbors and then attends to one token after every $w$ tokens in a strided manner. The sparsity pattern for the $k$-th token reads

$$
\begin{aligned}
\mathcal{A}_k^1 &= [n] \cap \{k - \lceil w/2 \rceil, \dots, k-1, k, k+1, \dots, k + \lfloor w/2 \rfloor\}, \\
\mathcal{A}_k^2 &= [n] \cap \{\dots, k - 2w, k - w, k + w, k + 2w, \dots\}.
\end{aligned}
\tag{4}
$$

In the FIXED pattern, we divide the token into segments of length $w$. A token in a segment has access to other tokens in the same segment, and then the last tokens of the other segments:

$$
\mathcal{A}_k^1 = [n] \cap \{\lceil k/w \rceil \cdot w - w + 1, \dots, \lceil k/w \rceil \cdot w\}, \quad \mathcal{A}_k^2 = [n] \cap (\{k\} \cup \{w, 2w, 3w, \dots\}). \tag{5}
$$

The STRIDED and FIXED patterns satisfy both Assumption 1 and 2 for all values of $w$. Specifically, Assumption 1.3 holds with $s = 2$, because any token can directly/indirectly access all the tokens in two hops. As for Assumption 1.2, the identity permutation $\gamma(i) = i$ suffices to satisfy the assumption for both patterns. By choosing $w = O(\sqrt{n})$, sparse Transformers with the STRIDED and FIXED patterns achieve universal approximation power with $O(n^{3/2})$ connections per attention layer.

Guo et al. [11] consider the STAR sparsity pattern where they add an auxiliary *relay token* that attends to all the tokens, and the other tokens attend only to $2w$ neighboring tokens and the relay token. There is only one sparsity pattern, so $p = 1$. The STAR sparsity pattern can be written as

$$
\mathcal{A}_k^1 = \{n\} \cup \big\{(i-1) \bmod (n-1) + 1 \mid i \in \{k-w, \dots, k+w\}\big\} \text{ for } k \in [n-1], \ \mathcal{A}_n^1 = [n], \tag{6}
$$

where $w \geq 1$. For any fixed $w$, this sparse Transformer has $O(n)$ connections per attention layer, and it satisfies both assumptions. Specifically, Assumption 1.2 is satisfied with the identity permutation, i.e., $\gamma(i) = (i)$ for $i \in [n]$. Since any token can access other tokens within two hops, Assumption 1.3 is satisfied with $s = 2$. This demonstrates that $O(n)$ connections per layer suffice for sparse attention layers to have universal approximation power. One can similarly check that the sliding window sparsity patterns with/without global attention, proposed in Longformer [2], also satisfy the assumptions with $O(n)$ connections. For the BIGBIRD sparsity pattern [34], it is also straightforward to check that a combination of its window attention and global attention satisfies Assumption 1 with $O(n)$ connections. We state this interesting observation as a corollary below.

**Corollary 2.** *There exist sparse Transformers with $O(n)$ connections per self-attention layer that are universal approximators in the sense of Theorem 1.*

Recall that another line of results that replaces softmax $\sigma_S$ with sparse variants $\rho$ [8, 9, 35] also fits into our formulation, with $p = 1$ and $\mathcal{A}_k^1 = [n]$. As we show in § B, these alternative $\rho$'s satisfy Assumption 2. Thus, by Theorem 1, these models also have the universal approximation property.

## 4 Proof sketch and discussion

### 4.1 Sketch of proof of Theorem 1

Now, we sketch the proof of Theorem 1, which consists of three steps. Throughout the proof, we assume without loss of generality that $\mathbb{D} \subset [0, 1)^{d \times n}$.

**Step 1.** In the first step, we approximate $f \in \mathcal{F}$ with a piecewise constant function. Towards this, consider a class of piecewise constant functions $\overline{\mathcal{F}}(\delta)$ that map $\mathbb{D}$ to $\mathbb{R}^{d \times n}$, where $\delta > 0$ and $\delta^{-1}$ is an

integer. Any function in $\overline{\mathcal{F}}(\delta)$ maps cubes of the form $\boldsymbol{G} + [0, \delta)^{d \times n}$ to matrices $\boldsymbol{A_G} \in \mathbb{R}^{d \times n}$, where $\boldsymbol{G} \in \{0, \delta, \dots, 1 - \delta\}^{d \times n}$. We approximate $f$ with a function $\overline{f} \in \overline{\mathcal{F}}(\delta)$ such that $\mathsf{d}_p(f, \overline{f}) \leq \epsilon/2$, by choosing small enough $\delta$. We defer the statement and the proof to § C of the supplementary material.

**Step 2.** We then approximate $\overline{f} \in \overline{\mathcal{F}}(\delta)$ with a sparse Transformer network with a slightly modified architecture. In this architecture, we replace ReLU in the feed-forward layer with any piecewise linear activation $\phi \in \Phi$, where $\Phi$ denotes the class of (possibly discontinuous) piecewise linear functions with three pieces. We also replace $\rho$ in the sparse attention layer with the hardmax $\sigma_{\mathrm{H}}$ operator. We refer to the function class represented by the modified sparse Transformer as $\overline{\mathcal{ST}}^{h,m,r}$. By a careful construction, Lemma 3 shows that any $\overline{f} \in \overline{\mathcal{F}}(\delta)$ can be *exactly* represented by the modified Transformer. To this end, we first carefully choose the positional embedding $\boldsymbol{E}$. We then quantize the inputs using feed-forward layers (Lemma 6), construct a *contextual mapping* using self-attention layers to map the quantized inputs to unique "ids" (Lemma 7), and then construct a *value mapping* with feed-forward layers to map the ids to desired output values (Lemma 8). See § D and § E in the supplementary material for details.

**Lemma 3.** *For any $\overline{f} \in \overline{\mathcal{F}}(\delta)$, there exists $\overline{g} \in \overline{\mathcal{ST}}^{2,1,1}$ such that $\overline{f}(\boldsymbol{X}) = \overline{g}(\boldsymbol{X})$ for all $\boldsymbol{X} \in \mathbb{D}$.*

**Step 3.** The final step is to approximate the function $\overline{g} \in \overline{\mathcal{ST}}^{2,1,1}$ with a sparse Transformer $g \in \mathcal{ST}^{2,1,4}$. This is done by approximating $\phi$ and $\sigma_{\mathrm{H}}$ with ReLU and $\rho$, respectively, while carefully bounding the accumulation of errors introduced by the approximation. See § F in the supplementary material for the details.

**Lemma 4.** *For $\overline{g} \in \overline{\mathcal{ST}}^{2,1,1}$ in Lemma 3, there exists $g \in \mathcal{ST}^{2,1,4}$ such that $\mathsf{d}_p(\overline{g}, g) \leq \epsilon/2$.*

Combining these three steps, we establish that $\mathsf{d}_p(f, g) \leq \mathsf{d}_p(f, \overline{f}) + \mathsf{d}_p(\overline{f}, \overline{g}) + \mathsf{d}_p(\overline{g}, g) \leq \epsilon$. $\quad\square$

**How many layers are sufficient?** In § D, Lemmas 6–8 show that we need $\frac{dn}{\delta}$ sparse Transformer blocks (2) for quantization, $\frac{p(n-1)}{\delta^d} + s$ for the contextual mapping, and $\frac{n}{\delta^{dn}}$ for the value mapping. Recall that $p$ is from (2), $s$ is from Assumption 1, and $\delta$ is from Step 1 above. In comparison, § C of [33] shows that the dense counterpart requires $\frac{dn}{\delta}$, $\frac{n}{\delta^d} + 1$, and $\frac{n}{\delta^{dn}}$ Transformer blocks (1) for the three corresponding lemmas. Note two observations: 1) The value mapping **dominates** the depth, and its depth requirements are **identical** for the two cases; and 2) For contextual mappings (where the attention layers are used), we need roughly $p$ **times more** layers for sparse models. Recall from § 3.4 that $p$ is usually a small constant. These observations mean that sparse Transformers can achieve universal approximation using depth of the **same order** in $d$, $n$ and $\delta$ as the dense Transformers.

## 4.2 Key challenges in the proof

While the high level outline of the proof is similar to the one for dense Transformers [33], the proof in [33] crucially relies on having *all* connections for computing attention in each layer, which we do not have in sparse Transformers. The sparsity in attention mechanism and the choice of general probability map $\rho$ pose nontrivial challenges in the proof. We highlight the key differences below.

Establishing the Step 2 of the dense result [33] relies on constructing a *contextual mapping* using attention layers. A contextual mapping is a function that maps tokens in different sequences to unique values, thereby allowing Transformers to distinguish the same token appearing in different contexts. A crucial ingredient in the construction of such a mapping is a shift operation implemented with two attention heads in an attention layer. This shift operation involves each token taking the maximum and minimum over the entire sequence, which obviously cannot be done with sparse Transformers as it would require each token to attend to all the other tokens in the sequence. We circumvent this issue by carefully choosing the positional embedding $\boldsymbol{E}$ dependent on $\gamma$ (cf. Assumption 1.2), and ensuring that a similar shift operation is applied in a desired order even under sparsity.

As the final phase of the contextual mapping in [33], a single attention layer shifts the entire sequence by the maximum over the sequence. Again, this cannot be directly implemented due to sparsity. Using Assumption 1.3, we instead prove that by stacking $s$ sparse layers, one can successfully implement a similar operation that shifts the entire sequence by the maximum over the whole sequence, up to some controlled errors. This way, we overcome the difficulties posed by the sparsity and construct a new version of contextual mappings. The details can be found in § E.2 of the supplementary material.

Moreover, the proof of Step 3 in [33] uses the simple fact that softmax can approximate hardmax arbitrarily closely. Since we do not restrict ourselves to softmax and generalize the probability map,

**Table 1:** Accuracy on the synthetic copying task. Percentages in parentheses mark the sparsity levels.

| Depth | STRIDED | | | FIXED | | | STAR | RANDOM |
|---|---|---|---|---|---|---|---|---|
| | UNION (87%) | MULTIHEAD (93%) | SEQUENTIAL (93%) | UNION (87%) | MULTIHEAD (93%) | SEQUENTIAL (93%) | (87%) | (90%) |
| 1-layer | 0.82% | 0.82% | 0.80% | 7.04% | 0.76% | 0.80% | 1.53% | 33.14% |
| 2-layer | 100.00% | 100.00% | 81.24% | 69.26% | 56.45% | 96.01% | 29.70% | 63.41% |
| 3-layer | 100.00% | 100.00% | 100.00% | 99.98% | 99.08% | 98.58% | 42.18% | 70.29% |
| 4-layer | 100.00% | 100.00% | 100.00% | 100.00% | 99.64% | 100.00% | 83.57% | 95.49% |

a more careful argument is required. Since there are many layers in the network $\overline{g}$, it turns out that approximating it with an original sparse Transformer in $\mathcal{ST}^{2,1,4}$ requires carefully controlling the approximation errors accumulated over layers. The proof of Lemma 4 in § F of the supplementary material shows that this is indeed possible by utilizing Assumption 2.

## 5 Experiments

We now present our experimental study comparing different design and implementation choices, including sparsity patterns and levels, on four tasks: i) a synthetic copying task, ii) language modeling, iii) translation, and iv) GLUE tasks. Our goal is to understand the effect of such choices while employing sparse Transformers to the tasks with small sequence lengths, complementing the existing results for sparse Transformers on long sequence tasks.

### 5.1 Experiment Settings

We consider four sparsity patterns: STRIDED (4), FIXED (5), STAR (6) and RANDOM. The first three patterns are proposed in [6] and [11]; we test them for different values of $w$. In case of the RANDOM pattern, given a sparsity level, we make connections uniformly at random. Following [6], STRIDED and FIXED patterns are tested for three different head configurations: i) SEQUENTIAL, where the sparse attention layers alternate between $\{\mathcal{A}_k^1\}$ and $\{\mathcal{A}_k^2\}$, as described in the previous sections; ii) UNION, where all sparse attention layers use the sparsity pattern $\{\mathcal{A}_k^1 \cup \mathcal{A}_k^2\}$; and iii) MULTIHEAD, where half of the attention heads in every attention layer use $\{\mathcal{A}_k^1\}$ and the other half use $\{\mathcal{A}_k^2\}$. Note that, given the same sequence length, UNION is less sparse than the other two configurations. Thus, to ensure fair comparisons, we compare different configurations based on their sparsity levels.

We use maximum sequence length 256 in all our experiments, except 128 for GLUE tasks. For the copying task, we experiment with only one sparse Transformer block (cf. Eq (2)), with varying numbers of attention layers with 4 attention heads. For language modeling and translation, we use the Tensor2Tensor [29] framework and employ 12-block and 6-block (respectively) Transformers with 8 attention heads per block. For GLUE tasks, we experiment with the BERT$_{\text{BASE}}$ model. For more details of the setup, see § G of the supplementary material.

### 5.2 Results

**Copying task.** We consider a synthetic copying task proposed in [14], where the input sequence has the format 0s0s, where s is a 127 length sequence of symbols in $[0, 127]$. The models have to predict (copy) the second part, given the first half of the input. This task tests the ability of sparse Transformers to communicate the information. Table 1 presents the results for this task. Except for the STAR and RANDOM patterns, we can see that the networks learn to copy the sequences with four sparse attention layers. One possible explanation for the bad performance of STAR is that, except for the relay token, it only attends to local neighbors while the task requires to copy distant tokens.

**Language modeling.** We conduct the language modeling experiments on the One Billion Word Benchmark [5] which has almost one billion tokens and a vocabulary of more than 800K unique tokens. In Figure 1a, we plot the perplexity against the sparsity level. We observe that the STRIDED pattern and the STAR achieve the best performance across all sparsity levels. For both the STRIDED and FIXED patterns, the UNION configuration shows the best performance.

**Translation.** For the translation task, we train the model on WMT18 English-Czech (en-cs) dataset and test it on the Newstest 2015 dataset. We plot the BLEU score against the sparsity level in Figure 1b. We apply the same sparsity pattern to both the encoder and the decoder. The STRIDED

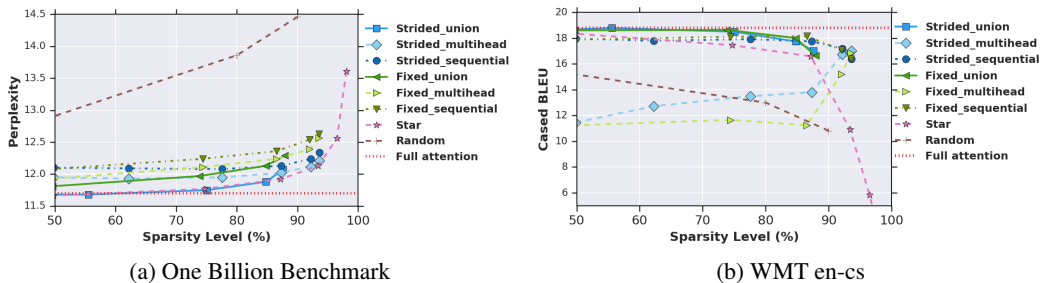

**Figure 1.** Comparison of sparsity patterns and different head configurations on the One Billion Benchmark (a language modeling task) and WMT en-cs (a translation task). Note that the number of connections in the attention layers goes down as we increase the sparsity level.

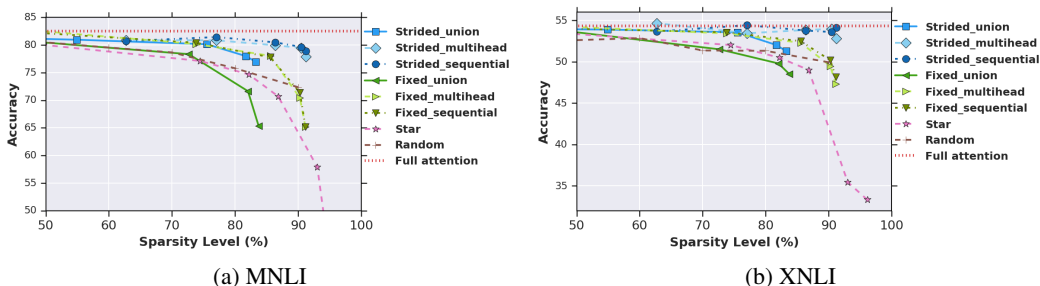

**Figure 2.** Comparison of sparsity patterns and different head configurations on the MNLI and XNLI (sentence-pair classification tasks), using the BERT_BASE model.

and FIXED patterns with UNION configuration show the best scores, which are similar to the dense attention. The UNION configuration is also the least sensitive to the sparsity levels.

**GLUE Tasks.** We experiment with the BERT_BASE model and report results on two sentence-pair classification tasks: MNLI [30] (Figure 2a) and XNLI [7] (Figure 2b). We plot the average accuracy of three runs on the dev set against the sparsity level. Additional results of the CoLA and MRPC tasks are reported in § H of the supplementary material.

**Discussion.** In all tasks, the RANDOM pattern performs worse than the deterministic patterns, demonstrating the need for a careful design of sparsity patterns. Overall, our experiments suggest that the design of the optimal sparsity patterns is heavily dependent on specific tasks. For example, the STAR pattern shows the best performance on the language modeling task, while having trouble with copying, translation, and BERT experiments. Among the three head configurations tested for STRIDED and FIXED, the UNION performs the best in language modeling and translation but suffers in BERT tasks. In translation experiments, we see an interesting trend that the performance of MULTIHEAD configuration improves as sparsity increases. We conjecture that this is due to the fact that in STRIDED and FIXED, we have $|\mathcal{A}_k^1| = O(w)$ and $|\mathcal{A}_k^2| = O(n/w)$ (cf. Eqs (4) and (5)), so the sparsest choice of $w = O(\sqrt{n})$ is the one with the best "balance" between $|\mathcal{A}_k^1|$ and $|\mathcal{A}_k^2|$.

## 6 Conclusion

Recently, sparse Transformers have received a lot of attention as they enable more efficient/faster attention mechanisms for the tasks with very long sequence lengths. We take an initial step to provide a theoretical understanding of these models. We provide a unifying framework that captures existing sparse attention models, and prove a universal approximation theorem for sparse Transformers which holds under intuitive conditions on sparsity patterns and probability maps. We also carry out experiments comparing different sparsity patterns and levels on standard NLP tasks. We hope that this work will shed light on the understanding of sparsity in attention layers, and provide guidance for the design of sparse attention models.

## Broader Impact

This work studies theoretical aspects of a class of widely used neural network models in NLP and related areas. Since we do not propose a new method nor a new dataset, we expect that the impact of this work on ethical aspects and future societal consequences will be small, if any. Other than that, this work brings new insights into the sparsity in attention models, hence may make an impact on the study of faster and more efficient NLP models.

## Acknowledgments and Disclosure of Funding

CY acknowledges partial support as a graduate Research Assistant from the NSF Grant (CAREER 1846088). CY also acknowledges Korea Foundation for Advanced Studies for their support.

## Footnotes

[1]Note that this corresponds to our Assumption 1.3.

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
