[Supplementary Material]

# A   Outline and notation

The supplementary material is organized as follows. First, § B proves that the softmax operator as well as its sparse versions indeed satisfy Assumption 2. Next, § C provides formal statements of Step 1 in the proof sketch (§ 4.1). The outline of proof of Lemma 3 (Step 2 in the proof sketch) is presented in § D, followed by a separate section (§ E) proving the three key sublemmas in the proof. The proof of Step 3, Lemma 4, is given in § F. Lastly, § G and § H present the detailed setup of our experiments and additional experiment results, respectively.

We next review some of the notation and also introduce additional notation used throughout the supplementary material. For a positive integer $a$, let $[a] := \{1, \ldots, a\}$. For $a, b, c \in \mathbb{R}$ where $b - a > 0$ is an integer multiple of $c > 0$, we write $[a : c : b] := \{a, a + c, a + 2c, \ldots, b - c, b\}$. For any matrix $\boldsymbol{A} \in \mathbb{R}^{d \times n}$, let $\boldsymbol{A}_j$ denote its $j$-th column, and $\boldsymbol{A}_\mathcal{S}$ denote the submatrix consisting of columns of $\boldsymbol{A}$ in the index set $\mathcal{S} \subseteq [n]$. We also use $A_{i,j}$ to denote its $(i, j)$-th entry. Let $\mathbb{1}\{\cdot\}$ be the 0-1 indicator for an event. Let $\boldsymbol{1}_n \in \mathbb{R}^n$ be a vector whose components are all 1.

# B   Sparse probability maps satisfy Assumption 2

In this section, we show that the softmax operator $\sigma_\mathrm{S}$ as well as the probability maps $\rho$ used to replace softmax in the existing approaches, namely softmax with only top-$k$ inputs [35], sparselin-gen [9], and $\alpha$-entmax [8], all satisfy Assumption 2. We restate the assumption for reader's convenience:

**Assumption 2.** *For any $\zeta > 0$ and $\eta \in (0, 1]$, $\exists\, t > 0$ such that, for any column input $\boldsymbol{v}$ satisfying $v_{j^*} - \max_{j \neq j^*} v_j \geq \zeta$ (where $j^* = \arg\max_j v_j$), we have $\rho[t\boldsymbol{v}]_{j^*} \geq 1 - \eta$ and $\sum_{j \neq j^*} \rho[t\boldsymbol{v}]_j \leq \eta$.*

As in the assumption, we only consider the operation of these probability maps on a single vector, as they are applied column-wise. For each of the probability maps, we will show that for any $\zeta > 0$ and $\eta \in (0, 1]$, we can choose $t > 0$ that satisfies the conditions of Assumption 2.

## B.1   Softmax & softmax with top-$k$ inputs

Given an input vector $\boldsymbol{v} \in \mathbb{R}^n$, the $j$-th coordinate of the output of softmax $\sigma_\mathrm{S}[\boldsymbol{v}]$ is defined as

$$\sigma_\mathrm{S}[\boldsymbol{v}]_j := \frac{\exp(v_j)}{\sum_{i=1}^n \exp(v_i)}.$$

We assume without loss of generality that the entry of $\boldsymbol{v}$ is in decreasing order, where the first two entries satisfy $v_1 - v_2 \geq \zeta$. For any such $\zeta > 0$ and any $0 < \eta \leq 1$, our aim is to show the existence of $t > 0$ such that $\sigma_\mathrm{S}[t\boldsymbol{v}]_1 = \frac{\exp(tv_1)}{\sum_{i=1}^n \exp(tv_i)} \geq 1 - \eta$. Then, $\sum_{j=2}^n \sigma_\mathrm{S}[t\boldsymbol{v}]_j \leq \eta$ follows.

Now, since $v_i \leq v_1 - \zeta$ for $i \in [2 : n]$, note that

$$\sigma_\mathrm{S}[t\boldsymbol{v}]_1 = \frac{\exp(tv_1)}{\sum_{i=1}^n \exp(tv_i)} \geq \frac{\exp(tv_1)}{\exp(tv_1) + (n-1)\exp(tv_1 - t\zeta)} = \frac{1}{1 + (n-1)\exp(-t\zeta)}.$$

Since $\frac{1}{1+(n-1)\exp(-t\zeta)}$ is an increasing function in $t > 0$, one can increase $t$ sufficiently large to make it greater than $1 - \eta$.

The same argument holds for the softmax with top-$k$ inputs, used in [35]. By the assumption on $\boldsymbol{v}$, entries $v_1, \ldots, v_k$ are the top $k$ components. Thus,

$$\rho[t\boldsymbol{v}]_1 \geq \frac{1}{1 + (k-1)\exp(-t\zeta)} \geq 1 - \eta$$

can be satisfied by choosing large enough $t > 0$.

## B.2   Sparselin-gen

We now consider the case where $\rho$ is sparselin-gen [15], which was used to sparsify the attention score matrices in [9]. Given a regularization parameter $\lambda \in [0, 1)$, the sparselin-gen used in [9] is defined as

$$\rho[\boldsymbol{v}] := \arg\min_{\boldsymbol{p} \in \Delta^{n-1}} \|\boldsymbol{p} - \boldsymbol{v}\|^2 - \lambda \|\boldsymbol{p}\|^2,$$

where $\Delta^{n-1} := \{\boldsymbol{p} \in \mathbb{R}^n \mid \boldsymbol{p} \geq \boldsymbol{0}, \sum_{i=1}^n p_i = 1\}$ is the probability simplex. Then, the solution for optimization problem above can be written as

$$\rho[\boldsymbol{v}]_j = \max\left\{0, \frac{v_j - \tau(\boldsymbol{v})}{1 - \lambda}\right\}, \text{ for } j \in [n],$$

where $\tau : \mathbb{R}^n \to \mathbb{R}$ is a threshold function that chooses the threshold $\tau(\boldsymbol{v})$ such that $\sum_{j=1}^n \rho[\boldsymbol{v}]_j = 1$.

Now, assume without loss of generality that the entry of $\boldsymbol{v}$ is in decreasing order, where the first two entries satisfy $v_1 - v_2 \geq \zeta$. For any such $\zeta > 0$ and any $0 < \eta \leq 1$, our aim is to show the existence of $t > 0$ such that $\rho[t\boldsymbol{v}]_1 \geq 1 - \eta$. This is done by choosing $t = \frac{1-\eta}{\zeta}$. To see this, notice that if $v_j$'s are in decreasing order, then $\rho[\boldsymbol{v}]_j$ are also in decreasing order. Now consider

$$\rho[t\boldsymbol{v}]_1 = \max\left\{0, \frac{tv_1 - \tau(t\boldsymbol{v})}{1 - \lambda}\right\}, \quad \rho[t\boldsymbol{v}]_2 = \max\left\{0, \frac{tv_2 - \tau(t\boldsymbol{v})}{1 - \lambda}\right\}.$$

If $\rho[t\boldsymbol{v}]_2 = 0$, then $\rho[t\boldsymbol{v}]_j = 0$ for all $j = 3, \ldots, n$, and $\rho[t\boldsymbol{v}]_1 = 1 \geq 1 - \eta$. If $\rho[t\boldsymbol{v}]_2 > 0$, then

$$\rho[t\boldsymbol{v}]_1 - \rho[t\boldsymbol{v}]_2 = \frac{tv_1 - \tau(t\boldsymbol{v})}{1 - \lambda} - \frac{tv_2 - \tau(t\boldsymbol{v})}{1 - \lambda} = \frac{t(v_1 - v_2)}{1 - \lambda} \geq t(v_1 - v_2) \geq t\zeta = 1 - \eta.$$

### B.3 $\alpha$-entmax

Next, we consider the case where $\rho$ is $\alpha$-entmax [21], which was used to sparsify the attention score matrices in [8]. Given a parameter $\alpha \geq 1$, the $\alpha$-entmax is defined as

$$\rho[\boldsymbol{v}] := \arg\max_{\boldsymbol{p} \in \Delta^{n-1}} \boldsymbol{p}^T \boldsymbol{v} + H_\alpha(\boldsymbol{v}),$$

where $\Delta^{n-1}$ is the probability simplex and $H_\alpha$ is the Tsallis continuous family of entropies

$$H_\alpha(\boldsymbol{v}) := \begin{cases} \frac{1}{\alpha(\alpha-1)} \sum_j v_j - v_j^\alpha & \alpha > 1, \\ -\sum_j v_j \log v_j & \alpha = 1. \end{cases}$$

As shown in [8], the solution of $\alpha$-entmax is equal to softmax if $\alpha = 1$, and otherwise ($\alpha > 1$) it is given in the form

$$\rho[\boldsymbol{v}]_j = \left[\max\{0, (\alpha - 1)v_j - \tau(\boldsymbol{v})\}\right]^{\frac{1}{\alpha-1}}, \text{ for } j \in [n],$$

where $\tau : \mathbb{R}^n \to \mathbb{R}$ is a threshold function that chooses the threshold $\tau(\boldsymbol{v})$ such that $\sum_{j=1}^n \rho[\boldsymbol{v}]_j = 1$. Since softmax ($\alpha = 1$) is already covered above, we focus on $\alpha > 1$.

Again, assume without loss of generality that the entry of $\boldsymbol{v}$ is in decreasing order, where the first two entries satisfy $v_1 - v_2 \geq \zeta$. For any such $\zeta > 0$ and any $0 < \eta \leq 1$, our aim is to show the existence of $t > 0$ such that $\rho[t\boldsymbol{v}]_1 \geq 1 - \eta$. This is done by choosing $t = 1/\zeta(\alpha-1)$.

Note that $(\alpha - 1)t(v_1 - v_2) \geq 1$ due to our choice of $t$. Then, we will show that with such a $t$, $\rho[t\boldsymbol{v}]_1 = 1$ must hold. For the sake of contradiction, suppose not: $\rho[t\boldsymbol{v}]_1 < 1$. Then, by monotonicity of $\rho[t\boldsymbol{v}]_j$, we have $\rho[t\boldsymbol{v}]_2 > 0$. This means

$$\rho[t\boldsymbol{v}]_2 = \left[(\alpha - 1)tv_2 - \tau(t\boldsymbol{v})\right]^{\frac{1}{\alpha-1}} > 0,$$

in particular, we have $(\alpha - 1)tv_2 - \tau(t\boldsymbol{v}) > 0$. However, recall that $(\alpha - 1)t(v_1 - v_2) \geq 1$, which implies $(\alpha - 1)tv_1 - \tau(t\boldsymbol{v}) > 1$. This results in

$$\rho[t\boldsymbol{v}]_1 = \left[(\alpha - 1)tv_1 - \tau(t\boldsymbol{v})\right]^{\frac{1}{\alpha-1}} > 1,$$

thus contradicting $\rho[t\boldsymbol{v}]_1 < 1$. Therefore, $\rho[t\boldsymbol{v}]_1 = 1$ must hold.

## C  Details of the Step 1 in the proof sketch (§ 4.1)

We start by formally defining the function class $\overline{\mathcal{F}}(\delta)$.

$$\overline{\mathcal{F}}(\delta) := \left\{ \boldsymbol{Z} \mapsto \sum_{\boldsymbol{G} \in \mathbb{G}_\delta} \boldsymbol{A_G} \mathbb{1}\left\{\boldsymbol{Z} \in \boldsymbol{G} + [0, \delta)^{d \times n}\right\} \mid \boldsymbol{Z} \in \mathbb{D}, \boldsymbol{A_G} \in \mathbb{R}^{d \times n} \right\},$$

where $\mathbb{G}_\delta := \{0, \delta, \ldots, 1 - \delta\}^{d \times n}$. We now state and prove the lemma.

**Lemma 5.** *For any $f \in \mathcal{F}$ and $\epsilon > 0$, there exists a small enough $\delta > 0$ such that there exists $\overline{f} \in \overline{\mathcal{F}}(\delta)$ such that $\mathsf{d}_p(f, \overline{f}) \leq \epsilon/2$.*

**Proof** Since $f : \mathbb{D} \to \mathbb{R}^{d \times n}$ is a continuous function on a compact domain, it is uniformly continuous. Also, continuity is defined with respect to entry-wise $\ell_p$ norm which is equivalent to entry-wise $\ell_\infty$ norm, uniform continuity leads to

$$\forall \epsilon > 0, \exists \delta > 0 \text{ such that } \forall \boldsymbol{X}, \boldsymbol{Y}, \|\boldsymbol{X} - \boldsymbol{Y}\|_\infty < \delta \implies \|f(\boldsymbol{X}) - f(\boldsymbol{Y})\|_p < \epsilon/2.$$

Then, suppose we create a set of cube grid points $\mathbb{G}_\delta := \{0, \delta, \ldots, 1-\delta\}^{d \times n}$, and define a piece-wise constant approximation

$$\overline{f}(\boldsymbol{X}) = \sum\nolimits_{\boldsymbol{G} \in \mathbb{G}_\delta} f(\boldsymbol{G}) \mathbb{1} \left\{ \boldsymbol{X} \in \boldsymbol{G} + [0, \delta)^{d \times n} \right\}.$$

Note that for any $\boldsymbol{X} \in \boldsymbol{G} + [0, \delta)^{d \times n}$ we have $\|\boldsymbol{X} - \boldsymbol{G}\|_\infty < \delta$, so we have

$$\left\| f(\boldsymbol{X}) - \overline{f}(\boldsymbol{X}) \right\|_p = \|f(\boldsymbol{X}) - f(\boldsymbol{G})\|_p < \epsilon/2.$$

This implies that

$$\mathsf{d}_p(f, \overline{f}) = \left( \int_{\mathbb{D}} \left\| f(\boldsymbol{X}) - \overline{f}(\boldsymbol{X}) \right\|_p^p \right)^{1/p} \leq \epsilon/2,$$

finishing the proof of the lemma. $\qquad\square$

# D  Proof of Lemma 3 (Step 2 in § 4.1)

In this section, we describe in further details how modified sparse Transformers (the class $\overline{\mathcal{ST}}^{2,1,1}$) are able to exactly express arbitrary piecewise constant functions in $\overline{\mathcal{F}}(\delta)$. We show that we can compute a *contextual mapping* of the entire input sequences without relying on dense self-attention layers. The token-wise feed-forward layers then transform these contextual mappings to the desired output sequence.

To give a high level summary of the proof, we want to show that given a piece-wise constant function $\overline{f} \in \overline{\mathcal{F}}(\delta)$, there exists a modified Transformer network $\overline{g} \in \overline{\mathcal{ST}}^{2,1,1}$ that exactly represents $\overline{f}$. Recall first that the function class $\overline{\mathcal{ST}}^{2,1,1}$ has an additive positional embedding matrix $\boldsymbol{E} \in \mathbb{R}^{d \times n}$ that is added to input before the input is fed to the network. We start by choosing the positional embedding $\boldsymbol{E}$ and construct a Transformer network that implements quantization of the input, contextual mapping of the quantized input, and value mapping of the context ids.

1. Choose the positional embedding $\boldsymbol{E}$ according to $\gamma$ in Assumption 1.2. After addition, each column of the input $\boldsymbol{X}_k + \boldsymbol{E}_k$ are in disjoint intervals.

2. Given the input $\boldsymbol{X} + \boldsymbol{E}$, a series of modified feed-forward layers quantizes it so that each entry of the quantized input has a value in $\{0, \delta, \ldots, n - \delta\}$ (Lemma 6).

3. Next, a series of modified sparse self-attention layers takes the quantized input $\boldsymbol{H}$ and implement a *contextual mapping* $q$ such that, for different quantized input sequences $\boldsymbol{H}$ and $\boldsymbol{H}'$, all the elements in $q(\boldsymbol{H})$ and $q(\boldsymbol{H}')$ are distinct (Lemma 7).

4. Finally, a series of modified feed-forward layers maps each element in the context id $q(\boldsymbol{H})$ to the desired output value of $\overline{f} \in \overline{\mathcal{F}}$ at the input $\boldsymbol{X}$ (Lemma 8).

We defer the proofs of Lemmas 6, 7, and 8 to a separate section: see § E.

Before discussing the details of each step, we note that although a Transformer network stacks self-attention and feed-forward layers in an alternate manner, we can use a series of arbitrary number of the same layers, thanks to skip connections. The outline of the proof is similar to [33], but key component in their proof called selective shift operation relies on the fact that each token can attend to the entire sequence; this is not true in sparse Transformers, which poses a nontrivial challenge. We overcome this issue by a more careful construction of the positional embedding $\boldsymbol{E}$ and sparse self-attention layers.

### D.1 Choosing the positional embedding

Recall from Assumption 1.2 that there exists a permutation $\gamma : [n] \to [n]$ such that for all $i \in [n-1]$, $\gamma(i)$ is one of the tokens that the $\gamma(i+1)$-th token directly attends to. Using this permutation $\gamma$, we choose the columns of positional embedding $\boldsymbol{E}$ in the following way:

$$\boldsymbol{E}_{\gamma(1)} = (n-1)\mathbf{1}_n, \text{ and } \boldsymbol{E}_{\gamma(i)} = (i-2)\mathbf{1}_n, \text{ for } i \in [2:n]$$

As a result, the $\gamma(1)$-th column of $\boldsymbol{X}+\boldsymbol{E}$ will be in the range $[n-1,n)^d$, and similarly $\boldsymbol{X}_{\gamma(i)}+\boldsymbol{E}_{\gamma(i)} \in [i-2,i-1)^d$ for $i \in [2:n]$. This means that the entries corresponding to different tokens lie be in disjoint intervals of the form $[j, j+1)$, where $j \in [0:n-1]$.

### D.2 Quantization by feed-forward layers

Note from the previous step that each entry of $\boldsymbol{X} + \boldsymbol{E}$ must be in $[0,n)$. Next, we quantize this interval $[0,n)$ of input using to a set of $\delta$-grid points $\{0, \delta, \ldots, n-\delta\}$. This allows us to deal with finite set of values, which proves useful in the later stages of the proof. The next lemma shows that the quantization can be carried out using a seried of the modified feed-forward layers.

**Lemma 6.** *Consider a entry-wise quantization map $g_q^{\text{ent}} : \mathbb{R} \to \mathbb{R}$:*

$$g_q^{\text{ent}}(t) = \begin{cases} k\delta & \text{if } k\delta \le t < (k+1)\delta, \ \ k \in [0:n/\delta - 1], \\ t & \text{otherwise.} \end{cases}$$

*There exists a function $g_q : \mathbb{R}^{d \times n} \mapsto \mathbb{R}^{d \times n}$ composed of $\frac{dn}{\delta}$ token-wise feed-forward layers with $r = 1$ and an activation $\phi \in \Phi$, which implements the entry-wise quantization $g_q^{\text{ent}}$ to each entry of its input.*

### D.3 Contextual mapping by sparse self-attention layers

After the input $\boldsymbol{X} + \boldsymbol{E}$ is quantized, the output of $g_q$ must be in the following set $\mathbb{H}_\delta \subset \mathbb{R}^{d \times n}$:

$$\mathbb{H}_\delta := \{\boldsymbol{G} + \boldsymbol{E} \in \mathbb{R}^{d \times n} \mid \boldsymbol{G} \in \mathbb{G}_\delta\},$$

where $\mathbb{G}_\delta := \{0, \delta, \ldots, 1-\delta\}^{d \times n}$ was defined to be the $\delta$-cubic grid points of $[0,1)^{d \times n}$. Using this finite set of sequences, we construct a *contextual mapping* that maps each sequence in $\mathbb{H}_\delta$ to unique numbers. Recall that the sparse attention layer has $p$ sparsity patterns that rotate in cycles, and Assumption 1.3 assumes that one token directly/indirectly access all the other tokens after $s$ such sparse attention layers. We now state the lemma.

**Lemma 7.** *Assume that $n \ge 2$, and $\delta^{-1}$ is an integer satisfying $\delta^{-1} \ge 2$. Suppose that the sparse self-attention layers ($h = 2, m = 1$) satisfy Assumption 1 and employ the hardmax $\sigma_H$ operator, and that the positional embedding $\boldsymbol{E}$ was chosen as described in § D.1. Then, there exist a function $g_c : \mathbb{R}^{d \times n} \to \mathbb{R}^{d \times n}$ composed of $\frac{p(n-1)}{\delta^d} + s$ sparse self-attention layers, and a vector $\boldsymbol{u} \in \mathbb{R}^d$, such that $q(\boldsymbol{H}) := \boldsymbol{u}^T g_c(\boldsymbol{H})$ satisfies the following properties:*

1. *For any $\boldsymbol{H} \in \mathbb{H}_\delta$, the entries of $q(\boldsymbol{H})$ are all distinct.*
2. *For any $\boldsymbol{H}, \boldsymbol{H}' \in \mathbb{H}_\delta$ such that $\boldsymbol{H} \ne \boldsymbol{H}'$, all entries of $q(\boldsymbol{H})$, $q(\boldsymbol{H}')$ are distinct.*

This contextual mapping maps each unique sequence/context into different context ids, enabling the network to distinguish the same token appearing in different sequences.

### D.4 Value mapping by feed-forward layers

After the contextual mapping, we use the token-wise feed-forward layers to map each different context ids to the desired output value of the target function $\overline{f}$. More specifically, recall the function $g_c$ from Lemma 7. For any $\boldsymbol{H} \in \mathbb{H}_\delta$, we need to map the output $g_c(\boldsymbol{H})$ of Lemma 7 to the desired function value $\overline{f}(\boldsymbol{H} - \boldsymbol{E})$ (recall that $\boldsymbol{H}$ is the quantized input *after* adding $\boldsymbol{E}$ to $\boldsymbol{X}$, so we need to subtract $\boldsymbol{E}$). This is done by implementing a token-wise value mapping using the feed-forward layers.

**Lemma 8.** *There exists a function* $g_{\mathrm{v}} : \mathbb{R}^{d \times n} \to \mathbb{R}^{d \times n}$ *composed of* $n(\frac{1}{\delta})^{dn}$ *token-wise feed-forward layers* $(r = 1)$ *with an activation* $\phi' \in \Phi$ *such that* $g_{\mathrm{v}}$ *is defined by a token-wise function* $g_{\mathrm{v}}^{\mathrm{tkn}} : \mathbb{R}^d \to \mathbb{R}^d$ *on each column,*

$$g_{\mathrm{v}}(\boldsymbol{Z}) = \begin{bmatrix} g_{\mathrm{v}}^{\mathrm{tkn}}(\boldsymbol{Z}_1) & \cdots & g_{\mathrm{v}}^{\mathrm{tkn}}(\boldsymbol{Z}_n) \end{bmatrix},$$

*where for all* $\boldsymbol{H} \in \mathbb{H}_\delta$ *and* $k \in \{1, \ldots, n\}$,

$$g_{\mathrm{v}}^{\mathrm{tkn}}(g_{\mathrm{c}}(\boldsymbol{H})_k) = \overline{f}(\boldsymbol{H} - \boldsymbol{E})_k.$$

## D.5 Finishing the proof

Given Lemmas 6, 7, and 8, one can easily check that for any $\boldsymbol{G} \in \mathbb{G}_\delta := \{0, \delta, \ldots, 1 - \delta\}^{d \times n}$ and any input value $\boldsymbol{X} \in \boldsymbol{G} + [0, \delta)^{d \times n}$, we have

$$\begin{aligned}
g_{\mathrm{v}} \circ g_{\mathrm{c}} \circ g_{\mathrm{q}}(\boldsymbol{X} + \boldsymbol{E}) &= g_{\mathrm{v}} \circ g_{\mathrm{c}}(\boldsymbol{G} + \boldsymbol{E}) \\
&= \begin{bmatrix} g_{\mathrm{v}}^{\mathrm{tkn}}(g_{\mathrm{c}}(\boldsymbol{G} + \boldsymbol{E})_1) & g_{\mathrm{v}}^{\mathrm{tkn}}(g_{\mathrm{c}}(\boldsymbol{G} + \boldsymbol{E})_2) & \cdots & g_{\mathrm{v}}^{\mathrm{tkn}}(g_{\mathrm{c}}(\boldsymbol{G} + \boldsymbol{E})_n) \end{bmatrix} \\
&= \begin{bmatrix} \overline{f}(\boldsymbol{G})_1 & \overline{f}(\boldsymbol{G})_2 & \cdots & \overline{f}(\boldsymbol{G})_n \end{bmatrix} = \overline{f}(\boldsymbol{G}) = \overline{f}(\boldsymbol{X}).
\end{aligned}$$

Therefore, we have constructed a modified sparse Transformer network $\overline{g}(\boldsymbol{X}) := g_{\mathrm{v}} \circ g_{\mathrm{c}} \circ g_{\mathrm{q}}(\boldsymbol{X} + \boldsymbol{E})$ that satisfies $\overline{g}(\boldsymbol{X}) = \overline{f}(\boldsymbol{X})$ for all $\boldsymbol{X} \in \mathbb{D}$, hence proving Lemma 3.

# E  Proof of Lemmas 6, 7, and 8

## E.1 Proof of Lemma 6

The proof goes as follows. Using $\frac{n}{\delta}$ token-wise feed-forward layers, we implement the quantization function $g_{\mathrm{q}}^{\mathrm{ent}}$ that quantizes the first row of the input. Then we stack another $\frac{n}{\delta}$ layers to quantize the second row, and so on.

For the first row, we add $n/\delta$ layers of the following form, for $k \in [0 : n/\delta - 1]$.

$$\boldsymbol{Z} \mapsto \boldsymbol{Z} + \boldsymbol{e}^{(1)}\phi((\boldsymbol{e}^{(1)})^T \boldsymbol{Z} - k\delta \mathbf{1}_n^T), \quad \phi(t) = \begin{cases} 0 & t < 0 \text{ or } t \geq \delta, \\ -t & 0 \leq t < \delta, \end{cases}$$

where $\boldsymbol{e}^{(1)} \in \mathbb{R}^d$ is the first canonical basis vector $\boldsymbol{e}^{(1)} = (1, 0, \ldots, 0)$. Each layer quantizes $\boldsymbol{Z}_{1,:}$ in $[k\delta, k\delta + \delta)$ to $k\delta$, without modifying other intervals or other rows of $\boldsymbol{Z}$. Note that the activation $\phi$ is a piecewise linear function with three pieces; hence, $\phi \in \Phi$. Therefore, the layers satisfy the definition of modified feed-forward layers. We can now repeat the same construction for the $d - 1$ remaining rows.

## E.2 Proof of Lemma 7

In order to construct a network $g_{\mathrm{c}}$ that implements the contextual mapping, we first introduce two operations referred to as the *sparse selective shift operation* and *all-max-shift operation*, implemented by at most two (modified) sparse attention heads of head size 1. Then, we proceed to stack layers implementing the selective shift operations and all-max-shift operations, and prove that these layers map input $\boldsymbol{H} \in \mathbb{H}_\delta$ to unique context ids.

### E.2.1 Preliminaries

**Sparse selective shift operation.** Given any vector $\boldsymbol{u} \in \mathbb{R}^d$, first consider the following function implementable with a sparse attention head with head size 1 and sparsity pattern $\{\mathcal{A}_k^l\}_{k \in [n]}$. For $k \in [n]$, the function $\psi^l : \mathbb{R}^{d \times n} \to \mathbb{R}^{1 \times n}$ computes each of its output column in the following way:

$$\psi^l(\boldsymbol{Z}; b_Q)_k := \boldsymbol{u}^T \boldsymbol{Z}_{\mathcal{A}_k^l} \sigma_{\mathrm{H}}[(\boldsymbol{u}^T \boldsymbol{Z}_{\mathcal{A}_k^l})^T(\boldsymbol{u}^T \boldsymbol{Z}_k - b_Q)] = \begin{cases} \max_{j \in \mathcal{A}_k^l} \boldsymbol{u}^T \boldsymbol{Z}_j & \text{if } \boldsymbol{u}^T \boldsymbol{Z}_k > b_Q, \\ \min_{j \in \mathcal{A}_k^l} \boldsymbol{u}^T \boldsymbol{Z}_j & \text{if } \boldsymbol{u}^T \boldsymbol{Z}_k < b_Q. \end{cases}$$

One can consider a sparse self-attention layer that consists of two such heads, with $b_Q < b_Q'$:

$$\Psi^l(\boldsymbol{Z}; c, b_Q, b_Q') := \boldsymbol{Z} + \begin{bmatrix} c\boldsymbol{e}^{(1)} & -c\boldsymbol{e}^{(1)} \end{bmatrix} \begin{bmatrix} \psi^l(\boldsymbol{Z}; b_Q) \\ \psi^l(\boldsymbol{Z}; b_Q') \end{bmatrix}.$$

The $(1,k)$-th entry of $\Psi^l(\boldsymbol{Z}; c, b_Q, b'_Q)$ reads

$$\Psi^l(\boldsymbol{Z}; c, b_Q, b'_Q)_{1,k} = Z_{1,k} + c(\psi^l(\boldsymbol{Z}; b_Q)_k - \psi^l(\boldsymbol{Z}; b'_Q)_k)$$

$$= \begin{cases} Z_{1,k} + c(\max_{j \in \mathcal{A}^l_k} \boldsymbol{u}^T \boldsymbol{Z}_j - \min_{j \in \mathcal{A}^l_k} \boldsymbol{u}^T \boldsymbol{Z}_j) & \text{if } b_Q < \boldsymbol{u}^T \boldsymbol{Z}_k < b'_Q, \\ Z_{1,k} & \text{if } \boldsymbol{u}^T \boldsymbol{Z}_k \notin [b_Q, b'_Q]. \end{cases}$$

This means that for input columns $\boldsymbol{Z}_k$ satisfying $\boldsymbol{u}^T \boldsymbol{Z}_k \in (b_Q, b'_Q)$ *only*, $\Psi^l$ shifts up the first entry of $\boldsymbol{Z}_k$ by the difference of maximum and minimum values of $\boldsymbol{u}^T \boldsymbol{Z}_j$ over the sparsity pattern $j \in \mathcal{A}^l_k$, **while leaving other columns intact**. By choosing $b_Q$ and $b'_Q$ properly, we can selectively modify certain columns without touching other columns; we refer to this operation $\Psi^l$ as the *sparse selective shift operation*, and we will see later that this is indeed the key ingredient of our proof.

In fact, this operation is a sparse version of the selective shift operation used in [33]. Since $\mathcal{A}^l_k$ is usually only a small subset of $[n]$, one cannot calculate the maximum and minimum of $\boldsymbol{u}^T \boldsymbol{Z}_j$ over the whole sequence, as done in [33]. Instead, we use Assumption 1.2 and a more careful choice of $\boldsymbol{E}$ to get around the restriction posed by sparsity.

**All-max-shift operation.** Suppose the input $\boldsymbol{Z} \in \mathbb{R}^{d \times n}$ satisfies $\boldsymbol{u}^T \boldsymbol{Z} > 0$ entry-wise, for a vector $\boldsymbol{u} \in \mathbb{R}^d$. Then, the *all-max-shift operation* $\Omega^l : \mathbb{R}^{d \times n} \to \mathbb{R}^{d \times n}$ is a sparse self-attention layer that consists of one attention head:

$$\Omega^l(\boldsymbol{Z}; c) = \boldsymbol{Z} + c\boldsymbol{e}^{(1)}\psi^l(\boldsymbol{Z}; 0).$$

The $(1,k)$-th entry of $\Omega^l(\boldsymbol{Z}; c)$ reads

$$\Omega^l(\boldsymbol{Z}; c)_{1,k} = Z_{1,k} + c\psi^l(\boldsymbol{Z}; 0)_k = Z_{1,k} + c \max_{j \in \mathcal{A}^l_k} \boldsymbol{u}^T \boldsymbol{Z}_j.$$

So, for each column $k$, the all-max-shift operation shifts up the first entry of $\boldsymbol{Z}_k$ by the maximum value of $\boldsymbol{u}^T \boldsymbol{Z}_j$ over the sparsity pattern $j \in \mathcal{A}^l_k$. Unlike the selective shift operation, the all-max-shift operation is applied to all the columns.

**Column ids.** Recall that the any input to this step is in

$$\mathbb{H}_\delta := \{\boldsymbol{G} + \boldsymbol{E} \in \mathbb{R}^{d \times n} \mid \boldsymbol{G} \in \mathbb{G}_\delta := [0 : \delta : 1 - \delta]^{d \times n}\}.$$

Because of the way $\boldsymbol{E}$ is chosen according to the permutation $\gamma$ in Assumption 1.2, for any $\boldsymbol{H} \in \mathbb{H}_\delta$ we have

$$\boldsymbol{H}_{\gamma(1)} \in [n - 1 : \delta : n - \delta]^d,$$
$$\boldsymbol{H}_{\gamma(i)} \in [i - 2 : \delta : i - 1 - \delta]^d \text{ for all } i \in [2 : n].$$

Now consider $\boldsymbol{u} := (1, \delta^{-1}, \delta^{-2}, \ldots, \delta^{-d+1})$. It is easy to check that for any $\boldsymbol{H} \in \mathbb{H}_\delta$, the map $\boldsymbol{H}_k \mapsto \boldsymbol{u}^T \boldsymbol{H}_k$ is one-to-one, and

$$\boldsymbol{u}^T \boldsymbol{H}_{\gamma(1)} \in \left[(n-1)\sum_{i=0}^{d-1} \delta^{-i} : \delta : (n-1)\sum_{i=0}^{d-1} \delta^{-i} + \delta^{-d+1} - \delta\right],$$
$$\boldsymbol{u}^T \boldsymbol{H}_{\gamma(i)} \in \left[(i-2)\sum_{i=0}^{d-1} \delta^{-i} : \delta : (i-2)\sum_{i=0}^{d-1} \delta^{-i} + \delta^{-d+1} - \delta\right], \text{ for } i \in [2 : n]. \tag{7}$$

Hence, for each column $\boldsymbol{H}_k$, the inner product $\boldsymbol{u}^T \boldsymbol{H}_k$ is in an interval disjoint from the other columns. Thus, $\boldsymbol{u}^T \boldsymbol{H}_k$ can be thought as a "column id" that identifies the column's original input value $\boldsymbol{G}_k$ as well as its position $k$. Note furthermore that for any $\boldsymbol{H} \in \mathbb{H}_\delta$,

$$\boldsymbol{u}^T \boldsymbol{H}_{\gamma(2)} < \boldsymbol{u}^T \boldsymbol{H}_{\gamma(3)} < \cdots < \boldsymbol{u}^T \boldsymbol{H}_{\gamma(n)} < \boldsymbol{u}^T \boldsymbol{H}_{\gamma(1)}. \tag{8}$$

### E.2.2 Construction of layers

Given these preliminaries, we now describe our construction of $g_c$. Recall from Assumption 1.2 that the permutation $\gamma$ satisfies $\gamma(i - 1) \in \bigcup_{l=1}^p \mathcal{A}^l_{\gamma(i)}$ for $i \in [2 : n]$. From this, for $i \in [2 : n]$ we let

$l_i \in [p]$ be any index such that $\gamma(i-1) \in \mathcal{A}^{l_i}_{\gamma(i)}$. For simplicity of notation, let $z_k := \boldsymbol{u}^T \boldsymbol{H}_k$ for $k \in [n]$ and $\Delta = \sum_{i=0}^{d-1} \delta^{-i}$.

Next, starting from $i = 2$, we want to sequentially stack $\delta^{-d}$ sparse selective shift operations
$$\Psi^{l_i}(\cdot; \delta^{-d}, b - \delta/2, b + \delta/2),$$
in increasing order of $b \in \left[(i-2)\Delta : \delta : (i-2)\Delta + \delta^{-d+1} - \delta\right]$. That is, we want to add sparse attention layers with sparsity patterns $\mathcal{A}^{l_i}_{\gamma(i)}$ that apply the selective shift operation to each possible value of $z_{\gamma(i)}$. Recall that the sparsity patterns have to cycle from $\mathcal{A}^1_k$ to $\mathcal{A}^p_k$, so we have to place other remaining $p - 1$ sparsity patterns (whose indices are not $l_i$) in between the $\Psi^{l_i}$ layers. This can be done by setting all the other sparse attention layers to be the identity. This way, we stack a total of $p\delta^{-d}$ sparse attention layers for $i = 2$, another $p\delta^{-d}$ for $i = 3$, and so on, up to $i = n$.

After these layers, we further stack $s$ all-max-shift operations. For $i = 1, \ldots, s$, we add all-max-shift operations of the form
$$\Omega^{(i-1) \bmod p + 1}(\cdot; 2sn\delta^{-nd-1}).$$
Here, the superscript $(i-1) \bmod p + 1$ is there to make sure that we cycle through the sparsity patterns from 1 to $p$, until we stack $s$ layers in total. This finishes the construction of our function $g_c$ composed of $\frac{p(n-1)}{\delta^d} + s$ sparse self-attention layers.

### E.2.3 Selective shift operations

We now explain how these stacked self-attention layers implement a contextual mapping. This subsection will consider the selective shift operations part; all-max-shift operations are described in the next subsection. Suppose that after the input $\boldsymbol{H} \in \mathbb{H}_\delta$ is processed through the first $\frac{p(n-1)}{\delta^d}$ layers, we get $\widetilde{\boldsymbol{H}} \in \mathbb{R}^{d \times n}$ at the output. We will show at the end of this subsection that the map $\boldsymbol{H} \mapsto \boldsymbol{u}^T \widetilde{\boldsymbol{H}}_{\gamma(n)}$ is a one-to-one map for column $\gamma(n)$, so the selective shift operations compute a "unique id" for each possible input sequence $\boldsymbol{H} \in \mathbb{H}_\delta$.

**First selective shift.** First consider the first $p\delta^{-d}$ layers. Omitting layers that are identity, they are essentially selective shift operations $\Psi^{l_2}(\cdot; \delta^{-d}, b - \delta/2, b + \delta/2)$ for $b \in [0 : \delta : \delta^{-d+1} - \delta]$. Since $[0 : \delta : \delta^{-d+1} - \delta]$ is the set of possible values of $z_{\gamma(2)}$, these layers perform selective shift operation on the $\gamma(2)$-th column without changing the other columns. Each possible value of $\boldsymbol{H}_{\gamma(2)}$ undergoes one and only shift operation (by the corresponding layer with $b = \boldsymbol{u}^T \boldsymbol{H}_{\gamma(2)}$), by which the $(1, \gamma(2))$-th entry of the input is updated.

Recall by Assumption 1.2 that $\gamma(1) \in \mathcal{A}^{l_2}_{\gamma(2)}$, and that $z_{\gamma(1)}$ and $z_{\gamma(2)}$ are the maximum and minimum over the whole sequence $z_1, \ldots, z_n$ (see (8)). By Assumption 1.1 we also have $\gamma(2) \in \mathcal{A}^{l_2}_{\gamma(2)}$. Since both $\gamma(1)$ and $\gamma(2)$ are in $\mathcal{A}^{l_2}_{\gamma(2)}$, the maximum and minimum value of $z_j := \boldsymbol{u}^T \boldsymbol{H}_j$'s over $j \in \mathcal{A}^{l_2}_{\gamma(2)}$ are $z_{\gamma(1)}$ and $z_{\gamma(2)}$, respectively. Therefore, the $(1, \gamma(2))$-th entry of the input matrix is shifted up as follows:
$$\widetilde{H}_{1,\gamma(2)} := H_{1,\gamma(2)} + \delta^{-d}(z_{\gamma(1)} - z_{\gamma(2)}).$$
Let $\widetilde{\boldsymbol{H}}_{\gamma(2)}$ be the $\gamma(2)$-th column after the shift operation has shifted $H_{1,\gamma(2)}$ to $\widetilde{H}_{1,\gamma(2)}$. Then, define
$$\widetilde{z}_{\gamma(2)} := \boldsymbol{u}^T \widetilde{\boldsymbol{H}}_{\gamma(2)} = z_{\gamma(2)} + \delta^{-d}(z_{\gamma(1)} - z_{\gamma(2)}).$$
Note that $\widetilde{z}_{\gamma(2)} > z_{\gamma(1)}$ because
$$z_{\gamma(2)} + \delta^{-d}(z_{\gamma(1)} - z_{\gamma(2)}) > z_{\gamma(1)} \Leftrightarrow (\delta^{-d} - 1)(z_{\gamma(1)} - z_{\gamma(2)}) > 0,$$
which is true. Therefore, $\widetilde{z}_{\gamma(2)}$ becomes the new maximum among the current values $z_{\gamma(1)}, \widetilde{z}_{\gamma(2)}, z_{\gamma(3)}, \ldots, z_{\gamma(n)}$, and the new minimum element is $z_{\gamma(3)}$.

**Second selective shift.** We now consider the next $p\delta^{-d}$ layers, which are essentially $\Psi^{l_3}(\cdot; \delta^{-d}, b - \delta/2, b + \delta/2)$ for $b \in [\Delta : \delta : \Delta + \delta^{-d+1} - \delta]$. They apply the shift operation to the $\gamma(3)$-th column. Since we have $\gamma(2), \gamma(3) \in \mathcal{A}^{l_3}_{\gamma(3)}$, the shift operation similarly yields
$$\widetilde{z}_{\gamma(3)} := z_{\gamma(3)} + \delta^{-d}(\widetilde{z}_{\gamma(2)} - z_{\gamma(3)}) = z_{\gamma(3)} + \delta^{-d}(z_{\gamma(2)} - z_{\gamma(3)}) + \delta^{-2d}(z_{\gamma(1)} - z_{\gamma(2)}).$$

We can also show $\widetilde{z}_{\gamma(3)} > \widetilde{z}_{\gamma(2)}$, because

$$z_{\gamma(3)} + \delta^{-d}(\widetilde{z}_{\gamma(2)} - z_{\gamma(3)}) > \widetilde{z}_{\gamma(2)} \Leftrightarrow (\delta^{-d} - 1)(\widetilde{z}_{\gamma(2)} - z_{\gamma(3)}) > 0.$$

So after this operation $\widetilde{z}_{\gamma(3)}$ and $z_{\gamma(4)}$ are the new maximum and minimum over the updated sequence $z_{\gamma(1)}, \widetilde{z}_{\gamma(2)}, \widetilde{z}_{\gamma(3)}, z_{\gamma(4)}, \ldots, z_{\gamma(n)}$.

**Repeating the process.** The same process continues. The next $p\delta^{-d}$ layers shifts the $\gamma(4)$-th columns and results in $\widetilde{z}_{\gamma(4)}$ which is greater than $\widetilde{z}_{\gamma(3)}$. After the first $p(n-1)\delta^{-d}$ layers, all columns except $\gamma(1)$-th column have been shifted, resulting in $z_{\gamma(1)}, \widetilde{z}_{\gamma(2)}, \ldots, \widetilde{z}_{\gamma(n)}$ satisfying

$$(n-1)\Delta \le z_{\gamma(1)} < \widetilde{z}_{\gamma(2)} < \cdots < \widetilde{z}_{\gamma(n)}. \tag{9}$$

Let us denote the output of the $p(n-1)\delta^{-d}$-th layer as $\widetilde{H}$.

**Selective shifts implement a one-to-one map.** Next, we prove that the map from $H \in \mathbb{H}_\delta$ to

$$\widetilde{z}_{\gamma(n)} := \boldsymbol{u}^T \widetilde{H}_{\gamma(n)} = z_{\gamma(n)} + \sum_{i=1}^{n-1} \delta^{-id}(z_{\gamma(n-i)} - z_{\gamma(n+1-i)})$$

is one-to-one. Recall that for each column $H_k$, the map $H_k \mapsto \boldsymbol{u}^T H_k =: z_k$ is one-to-one. Also, permutation of columns is one-to-one, which implies that it suffices to show that the map $\begin{bmatrix} z_{\gamma(1)} & \cdots & z_{\gamma(n)} \end{bmatrix} \mapsto \widetilde{z}_{\gamma(n)}$ is one-to-one.

Suppose we have two sequences $\begin{bmatrix} z_{\gamma(1)} & \cdots & z_{\gamma(n)} \end{bmatrix}$ and $\begin{bmatrix} z'_{\gamma(1)} & \cdots & z'_{\gamma(n)} \end{bmatrix}$ that map to the same value of $\widetilde{z}_{\gamma(n)} = \widetilde{z}'_{\gamma(n)}$. Then,

$$0 = \widetilde{z}_{\gamma(n)} - \widetilde{z}'_{\gamma(n)} = z_{\gamma(n)} - z'_{\gamma(n)} + \sum_{i=1}^{n-1} \delta^{-id}(z_{\gamma(n-i)} - z_{\gamma(n+1-i)} - z'_{\gamma(n-i)} + z'_{\gamma(n+1-i)}).$$

Suppose $z_{\gamma(n)} \ne z'_{\gamma(n)}$. Since they both lie inside $[(n-2)\Delta : \delta : (n-2)\Delta + \delta^{-d+1} - \delta]$, we have

$$-\delta^{-d+1} + \delta \le z_{\gamma(n)} - z'_{\gamma(n)} \le \delta^{-d+1} - \delta.$$

Note that all the terms other than $z_{\gamma(n)} - z'_{\gamma(n)}$ are of "coarser resolution." For example, the first term

$$\delta^{-d}(z_{\gamma(n-1)} - z_{\gamma(n)} - z'_{\gamma(n-1)} + z'_{\gamma(n)})$$

in the summation can only take values $0, \delta^{-d+1}, -\delta^{-d+1}, 2\delta^{-d+1}, -2\delta^{-d+1}, \ldots$, so it can never cancel the difference $z_{\gamma(n)} - z'_{\gamma(n)}$ and make the sum $\widetilde{z}_{\gamma(n)} - \widetilde{z}'_{\gamma(n)}$ zero. This implies that $z_{\gamma(n)} = z'_{\gamma(n)}$ must hold.

Next, suppose $z_{\gamma(n-1)} \ne z'_{\gamma(n-1)}$. Since we have $z_{\gamma(n)} = z'_{\gamma(n)}$,

$$-\delta^{-2d+1} < \delta^{-d}(z_{\gamma(n-1)} - z_{\gamma(n)} - z'_{\gamma(n-1)} + z'_{\gamma(n)}) = \delta^{-d}(z_{\gamma(n-1)} - z'_{\gamma(n-1)}) < \delta^{-2d+1}.$$

But similarly, any other terms in the summation have coarser resolution than $\delta^{-2d+1}$, so they cannot cancel the difference $\delta^{-d}(z_{\gamma(n-1)} - z'_{\gamma(n-1)})$. Thus $z_{\gamma(n-1)} = z'_{\gamma(n-1)}$ must hold. Repeating the same argument up to $\gamma(1)$ proves that the two sequences must be equal: $\begin{bmatrix} z_{\gamma(1)} & \cdots & z_{\gamma(n)} \end{bmatrix} = \begin{bmatrix} z'_{\gamma(1)} & \cdots & z'_{\gamma(n)} \end{bmatrix}$. This proves that the map $H \mapsto \widetilde{z}_{\gamma(n)}$ is one-to-one and $\widetilde{z}_{\gamma(n)}$ can be seen as the unique id for the input sequence $H \in \mathbb{H}_\delta$.

### E.2.4 All-max-shift operations

Next, we explain the operation of the $s$ all-max-shift layers. Recall from Assumption 1.3 that any token can attend to all the other tokens after $s$ steps, either directly or indirectly. Also recall from the last subsection that the input to the first all-max-shift layer is $\widetilde{H}$, and the maximum entry of $\boldsymbol{u}^T \widetilde{H}$ is $\widetilde{z}_{\gamma(n)}$, the unique id for input $H$. From the statement of Lemma 7, the output after the $s$ all-max-shift operations for input $H$ is denoted as $g_c(H)$. In this subsection, we show that through $s$ all-max-shift operations, the maximum $\widetilde{z}_{\gamma(n)}$ will propagate to all tokens and be a "dominant" term, which determines the interval that $\boldsymbol{u}^T g_c(H)$ lies in. As a result, we can show Properties 7.1 and 7.2 of $g_c$ at the end.

**Some preliminaries.** Note that the unique id $\widetilde{z}_{\gamma(n)}$ has the following upper bound:

$$\widetilde{z}_{\gamma(n)} := z_{\gamma(n)} + \sum_{i=1}^{n-2} \delta^{-id}\big(z_{\gamma(n-i)} - z_{\gamma(n+1-i)}\big) + \delta^{-(n-1)d}\big(z_{\gamma(1)} - z_{\gamma(2)}\big)$$

$$\leq z_{\gamma(n)} + \delta^{-d} \sum_{i=1}^{n-2} \big(z_{\gamma(n-i)} - z_{\gamma(n+1-i)}\big) + \delta^{-(n-1)d}\big(z_{\gamma(1)} - z_{\gamma(2)}\big)$$

$$= z_{\gamma(n)} + \delta^{-d}\big(z_{\gamma(2)} - z_{\gamma(n)}\big) + \delta^{-(n-1)d}\big(z_{\gamma(1)} - z_{\gamma(2)}\big)$$

$$= \delta^{-(n-1)d} z_{\gamma(1)} - (\delta^{-(n-1)d} - \delta^{-d}) z_{\gamma(2)} - (\delta^{-d} - 1) z_{\gamma(n)}$$

$$\leq \delta^{-(n-1)d} z_{\gamma(1)} \leq \delta^{-(n-1)d}\big((n-1)\Delta + \delta^{-d+1} - \delta\big)$$

$$\leq \delta^{-(n-1)d}(n - 1 + \delta)(\delta^{-d} - 1) \leq \delta^{-nd} - \delta \tag{10}$$

where we used $\Delta := \sum_{i=0}^{d-1} \delta^{-i} = \frac{\delta^{-d}-1}{\delta^{-1}-1} \leq \delta^{-d} - 1$. A similar bound

$$\widetilde{z}_{\gamma(i)} \leq n\delta^{-id} - \delta \tag{11}$$

also holds from a similar derivation. Next, recall from Assumption 1.3 the definitions

$$\mathcal{S}_k^1 := \mathcal{A}_k^1, \;\; \mathcal{S}_k^t := \bigcup_{j \in \mathcal{A}_k^{(t-1) \bmod p+1}} \mathcal{S}_j^{t-1},$$

and that there exists $s \geq 1$ such that, for all $k \in [n]$, $\mathcal{S}_k^s = [n]$. Finally, the following inequality will be useful throughout: for any integer $s \geq 1$,

$$\left(\frac{2s+1}{2s}\right) \leq \left(\frac{2s+1}{2s}\right)^2 \leq \cdots \leq \left(\frac{2s+1}{2s}\right)^s \leq 2. \tag{12}$$

Let us now describe the operation that the all-max-shift layers $\Omega^{(i-1) \bmod p+1}(\cdot; 2sn\delta^{-nd-1})$, $i = 1, \ldots, s$, carry out.

**First all-max-shift.** The input to the first all-max-shift layer is $\widetilde{H}$. Let the output of the layer be $M^1$. Recall that $u^T \widetilde{H}$ consists of values $z_{\gamma(1)}, \widetilde{z}_{\gamma(2)}, \ldots, \widetilde{z}_{\gamma(n)}$, which are all strictly greater than 0 and strictly less than $n\delta^{-nd}$ (by (10)). So, for each column $k \in [n]$, the layer update reads

$$M_{1,k}^1 := \widetilde{H}_{1,k} + 2sn\delta^{-nd-1} \max_{j \in \mathcal{A}_k^1} u^T \widetilde{H}_j = \widetilde{H}_{1,k} + 2sn\delta^{-nd-1} u^T \widetilde{H}_{j_k^1},$$

where $j_k^1 := \arg\max_{j \in \mathcal{A}_k^1} u^T \widetilde{H}_j$. After the update, $u^T M_k^1$ is "dominated" by $2sn\delta^{-nd-1} u^T \widetilde{H}_{j_k^1}$, meaning that for any $k, k' \in [n]$,

$$u^T \widetilde{H}_{j_k^1} < u^T \widetilde{H}_{j_{k'}^1} \implies u^T M_k < u^T M_{k'}.$$

This is because the minimum gap between different values of $u^T \widetilde{H}_{j_k^1}$ is at least $\delta$, and we have

$$u^T \widetilde{H}_k < n\delta^{-nd} < 2sn\delta^{-nd-1} \cdot \delta,$$

so if $u^T \widetilde{H}_{j_k^1} < u^T \widetilde{H}_{j_{k'}^1}$, that solely determines the order $u^T M_k < u^T M_{k'}$ because $u^T \widetilde{H}_k$ cannot reverse it. Also, by the definition of $j_k^1$, for any index set $\mathcal{B} \in [n]$ we have

$$\max_{i \in \mathcal{B}} u^T \widetilde{H}_{j_i^1} = \max_{j \in \bigcup_{i \in \mathcal{B}} \mathcal{A}_i^1} u^T \widetilde{H}_j. \tag{13}$$

If $s \geq 2$, we move on to the second layer.

**Second all-max-shift.** At the second all-max-shift, we have sparsity patterns $\mathcal{A}_k^{1 \bmod p+1}$. Let us the output of this layer as $M^2$. For each column $k \in [n]$, the layer update reads

$$M_{1,k}^2 := M_{1,k}^1 + 2sn\delta^{-nd-1} \max_{j \in \mathcal{A}_k^{1 \bmod p+1}} u^T M_j^1 = M_{1,k}^1 + 2sn\delta^{-nd-1} u^T M_{j_k^2}^1,$$

where $j_k^2 := \arg\max_{j \in \mathcal{A}_k^{1 \bmod p+1}} \boldsymbol{u}^T \boldsymbol{M}_j^1$. If we look at the update more closely, we can apply (13) and get

$$\boldsymbol{u}^T \boldsymbol{M}_k^2 = \boldsymbol{u}^T \widetilde{\boldsymbol{H}}_k + 2sn\delta^{-nd-1}\boldsymbol{u}^T \widetilde{\boldsymbol{H}}_{j_k^1} + 2sn\delta^{-nd-1}(\boldsymbol{u}^T \widetilde{\boldsymbol{H}}_{j_k^2} + 2sn\delta^{-nd-1} \max_{i \in \mathcal{A}_k^{1 \bmod p+1}} \boldsymbol{u}^T \widetilde{\boldsymbol{H}}_{j_i^1})$$

$$= \boldsymbol{u}^T \widetilde{\boldsymbol{H}}_k + 2sn\delta^{-nd-1}(\boldsymbol{u}^T \widetilde{\boldsymbol{H}}_{j_k^1} + \boldsymbol{u}^T \widetilde{\boldsymbol{H}}_{j_k^2}) + (2sn\delta^{-nd-1})^2 \max_{j \in \mathcal{S}_k^2} \boldsymbol{u}^T \widetilde{\boldsymbol{H}}_j.$$

Again, the last term dominates the rest of the terms in $\boldsymbol{u}^T \boldsymbol{M}_k^2$, because the minimum gap between different values of $\max_{j \in \mathcal{S}_k^2} \boldsymbol{u}^T \widetilde{\boldsymbol{H}}_j$ is at least $\delta$, and

$$\boldsymbol{u}^T \boldsymbol{M}_k^2 - (2sn\delta^{-nd-1})^2 \max_{j \in \mathcal{S}_k^2} \boldsymbol{u}^T \widetilde{\boldsymbol{H}}_j = \boldsymbol{u}^T \widetilde{\boldsymbol{H}}_k + 2sn\delta^{-nd-1}(\boldsymbol{u}^T \widetilde{\boldsymbol{H}}_{j_k^1} + \boldsymbol{u}^T \widetilde{\boldsymbol{H}}_{j_k^2})$$

$$< (1 + 4sn\delta^{-nd-1})n\delta^{-nd} \le (1 + 4s)n^2\delta^{-2nd-1} \le (2sn\delta^{-nd-1})^2 \cdot \delta = 4s^2n^2\delta^{-2nd-1}.$$

The last inequality holds due to inequality (12), because

$$\left(\frac{2s+1}{2s}\right)^2 \le 2 \Leftrightarrow 1 + 4s \le 4s^2$$

is true for $s \ge 2$.

**Remaining all-max-shifts.** If $s \ge 3$, we move on to the third layer, which outputs $\boldsymbol{M}^3$. Similarly, we can show that $\boldsymbol{u}^T \boldsymbol{M}_k^3$ is dominated by $(2sn\delta^{-nd-1})^3 \max_{j \in \mathcal{S}_k^3} \boldsymbol{u}^T \widetilde{\boldsymbol{H}}_j$ because the rest of the terms in $\boldsymbol{u}^T \boldsymbol{M}_k^3$ is strictly upper-bounded

$$\boldsymbol{u}^T \boldsymbol{M}_k^3 - (2sn\delta^{-nd-1})^3 \max_{j \in \mathcal{S}_k^3} \boldsymbol{u}^T \widetilde{\boldsymbol{H}}_j < (1 + 3 \cdot 2sn\delta^{-nd-1} + 3 \cdot (2sn\delta^{-nd-1})^2)n\delta^{-nd-1},$$

which can then be shown to be smaller than $(2sn\delta^{-nd-1})^3 \cdot \delta$:

$$(1 + 3 \cdot 2sn\delta^{-nd-1} + 3 \cdot (2sn\delta^{-nd-1})^2)n\delta^{-nd} \le (1 + 6s + 12s^2)n^3\delta^{-3nd-2} \le 8s^3n^3\delta^{-3nd-3} \cdot \delta.$$

The last inequality is due to the fact that $1 + 6s + 12s^2 \le 8s^3$ for $s \ge 3$, which can derived from (12). Repeating this process, after all $s$ layers we get $\boldsymbol{M}^s$, and $\boldsymbol{u}^T \boldsymbol{M}_k^s$ is dominated by

$$(2sn\delta^{-nd-1})^s \max_{j \in \mathcal{S}_k^s} \boldsymbol{u}^T \widetilde{\boldsymbol{H}}_j = (2sn\delta^{-nd-1})^s \max_{j \in [n]} \boldsymbol{u}^T \widetilde{\boldsymbol{H}}_j = (2sn\delta^{-nd-1})^s \widetilde{z}_{\gamma(n)}.$$

This is because the remaining terms in $\boldsymbol{u}^T \boldsymbol{M}_k^s$ can be strictly upper-bounded

$$\boldsymbol{u}^T \boldsymbol{M}_k^s - (2sn\delta^{-nd-1})^s \widetilde{z}_{\gamma(n)} < \left(\sum_{i=0}^{s-1} \binom{s}{i}(2sn\delta^{-nd-1})^i\right) n\delta^{-nd},$$

which is then dominated by the smallest difference possible in $(2sn\delta^{-nd-1})^s \widetilde{z}_{\gamma(n)}$:

$$\left(\sum_{i=0}^{s-1} \binom{s}{i}(2sn\delta^{-nd-1})^i\right) n\delta^{-nd} \le \left(\sum_{i=0}^{s-1} \binom{s}{i}(2s)^i\right)(n\delta^{-nd-1})^{s-1}n\delta^{-nd}$$

$$= ((1 + 2s)^s - (2s)^s)(n\delta^{-nd-1})^s \cdot \delta \le (2sn\delta^{-nd-1})^s \cdot \delta.$$

The last inequality used $(1 + 2s)^s - (2s)^s \le (2s)^s$, derived from (12).

### E.2.5 Verifying Properties 7.1 and 7.2

After these all-max-shift operations, we define the output $\boldsymbol{M}^s$ of the last all-max-shift layers to be the output of the function $g_c$ for input $\boldsymbol{H}$, i.e., $g_c(\boldsymbol{H}) := \boldsymbol{M}^s$.

Property 7.1 requires that for any $\boldsymbol{H} \in \mathbb{H}_\delta$, all the components $\boldsymbol{u}^T g_c(\boldsymbol{H})$ need to be distinct. This is true, because for each column of $\boldsymbol{u}^T g_c(\boldsymbol{H})$, we have

$$\boldsymbol{u}^T g_c(\boldsymbol{H})_k \bmod 2sn\delta^{-nd} = \boldsymbol{u}^T \widetilde{\boldsymbol{H}}_k.$$

This is because anything added by the all-max-shift operations is an integer multiple of $2sn\delta^{-nd}$, and $\boldsymbol{u}^T\widetilde{\boldsymbol{H}}_k < n\delta^{-nd} < 2n\delta^{-nd}$ for all $k$. Recall that $\widetilde{\boldsymbol{H}}$ is the input matrix for the first max-shift operation, and that the components of $\boldsymbol{u}^T\widetilde{\boldsymbol{H}}$ are $z_{\gamma(1)}, \widetilde{z}_{\gamma(2)}, \ldots, \widetilde{z}_{\gamma(n)}$, which were shown to be distinct by (9). Since $\boldsymbol{u}^T g_c(\boldsymbol{H})_k$ produce distinct outputs for a $\mathrm{mod}$ operation, they themselves have to distinct. This proves Property 7.1.

Also, by the "domination" argument in the previous subsection, the output $g_c(\boldsymbol{H})$ has the property that for any column, $\boldsymbol{u}^T g_c(\boldsymbol{H})_k$ lies inside an interval determined by $\widetilde{z}_{\gamma(n)}$, the unique id for the input $\boldsymbol{H}$:

$$\boldsymbol{u}^T g_c(\boldsymbol{H})_k \in \left[ (2sn\delta^{-nd-1})^s \widetilde{z}_{\gamma(n)}, (2sn\delta^{-nd-1})^s (\widetilde{z}_{\gamma(n)} + \delta) \right),$$

and these intervals do not overlap because any different values of $\widetilde{z}_{\gamma(n)}$ must differ by at least $\delta$. This means that for any input $\boldsymbol{H}, \boldsymbol{H}' \in \mathbb{H}_\delta$, the components in $\boldsymbol{u}^T g_c(\boldsymbol{H})$ and $\boldsymbol{u}^T g_c(\boldsymbol{H}')$ lie in disjoint intervals. Together with Property 7.1, this proves Property 7.2.

### E.3 Proof of Lemma 8

To prove this lemma, we implement a token-wise function that maps

$$g_v^{\mathrm{tkn}}(g_c(\boldsymbol{H})_k) = \overline{f}(\boldsymbol{H} - \boldsymbol{E})_k,$$

for all $\boldsymbol{H} \in \mathbb{H}_\delta$ and $k \in [n]$. From the construction of Lemma 7, there are $n|\mathbb{H}_\delta| = \frac{n}{\delta^{dn}}$ distinct values of $\boldsymbol{u}^T g_c(\boldsymbol{H})_k$, and different values of $\boldsymbol{u}^T g_c(\boldsymbol{H})_k$ differ by at least $\delta$. The implementation of $g_v^{\mathrm{tkn}}$ can be done by stacking feed-forward layers so that each layer maps one unique number to the corresponding output column.

More precisely, choose any $\boldsymbol{H} \in \mathbb{H}_\delta$. For each of the $n$ values of $\boldsymbol{u}^T g_c(\boldsymbol{H})_k$, we add one feed-forward layer of the form

$$\boldsymbol{Z} \mapsto \boldsymbol{Z} + (\overline{f}(\boldsymbol{H} - \boldsymbol{E})_k - g_c(\boldsymbol{H})_k)\phi'(\boldsymbol{u}^T \boldsymbol{Z} - \boldsymbol{u}^T g_c(\boldsymbol{H})_k \mathbf{1}_n^T), \quad \phi'(t) = \begin{cases} 0 & t < -\delta/2 \text{ or } t \geq \delta/2, \\ 1 & -\delta/2 \leq t < \delta/2. \end{cases}$$

This layer updates any column $j$ of its input $\boldsymbol{Z}$ that satisfies $\boldsymbol{u}^T g_c(\boldsymbol{H})_k - \delta/2 \leq \boldsymbol{u}^T \boldsymbol{Z}_j < \boldsymbol{u}^T g_c(\boldsymbol{H})_k + \delta/2$, without modifying any other columns that are out of this range.

We stack these layers for all possible values of $\boldsymbol{H} \in \mathbb{H}_\delta$. After $\frac{n}{\delta^{dn}}$ such layers, we get the desired function $g_v$ that satisfies

$$g_v(\boldsymbol{Z}) = \begin{bmatrix} g_v^{\mathrm{tkn}}(\boldsymbol{Z}_1) & \cdots & g_v^{\mathrm{tkn}}(\boldsymbol{Z}_n) \end{bmatrix},$$

where for all $\boldsymbol{H} \in \mathbb{H}_\delta$ and $k \in [n]$,

$$g_v^{\mathrm{tkn}}(g_c(\boldsymbol{H})_k) = \overline{f}(\boldsymbol{H} - \boldsymbol{E})_k.$$

## F Proof of Lemma 4 (Step 3 in § 4.1)

In this section, we describe how the modified sparse Transformer network $\overline{g} \in \overline{\mathcal{ST}}^{2,1,1}$ constructed in Lemma 3 can be approximated with an original sparse Transformer network $g \in \mathcal{ST}^{2,1,4}$. Recall that $\overline{g}$ is a "modified" sparse Transformer network, which employ the hardmax $\sigma_H$ operators in place of $\rho$ operators in sparse self-attention layers and piecewise linear activations $\phi \in \Phi$ instead of ReLUs in feed-forward layers. The goal of this lemma is to approximate the function $\overline{g} = g_v \circ g_c \circ g_q \in \overline{\mathcal{ST}}^{2,1,1}$ with a standard sparse Transformer $g = \widetilde{g}_v \circ \widetilde{g}_c \circ \widetilde{g}_q \in \mathcal{ST}^{2,1,4}$ with accuracy $\mathsf{d}_p(\overline{g}, g) \leq \epsilon/2$. As the construction of $\overline{g}$ consists of three steps, we will approximate each of them step by step. The whole intuition behind the proof is that as long as we are considering $L_p$ approximation, we can approximate $\sigma_H$ and $\phi \in \Phi$ as closely as we want with $\rho$ and ReLUs, respectively. However, as the proof will show, controlling the aggregated error over layers is not a trivial job.

### F.1 Approximating the quantization function $g_q$ (Lemma 6)

We first consider approximating $g_q$ from Lemma 6 with a standard feed-forward layer counterpart, $\widetilde{g}_q$. Recall from § E.1 that the modified feed-forward layers used in $g_q$ are of the form

$$\boldsymbol{Z} \mapsto \boldsymbol{Z} + \boldsymbol{e}^{(i)}\phi((\boldsymbol{e}^{(i)})^T \boldsymbol{Z} - k\delta\mathbf{1}_n^T), \quad \phi(t) = \begin{cases} 0 & t < 0 \text{ or } t \geq \delta, \\ -t & 0 \leq t < \delta, \end{cases} \tag{14}$$

for $i \in [d]$ and $k \in [0 : n/\delta - 1]$. Note that the activation $\phi \in \Phi$ can be closely approximated by three ReLUs:

$$\widetilde{\phi}_\alpha(t) := -\mathrm{ReLU}(t) + \frac{1}{\alpha}\mathrm{ReLU}(t - (1-\alpha)\delta) - \frac{1-\alpha}{\alpha}\mathrm{ReLU}(t-\delta)$$

$$= \begin{cases} 0 & t \le 0 \text{ or } t \ge \delta, \\ -t & 0 \le t \le (1-\alpha)\delta, \\ \frac{1-\alpha}{\alpha}(t-\delta) & (1-\alpha)\delta \le t \le \delta, \end{cases}$$

where $0 < \alpha < 1$. Note that $\widetilde{\phi}_\alpha(t) = \phi(t)$ except for an interval $((1-\alpha)\delta, \delta)$, and by shrinking $\alpha > 0$ this interval can be made arbitrarily small. Consider approximating the layers (14) with standard feed-forward layers, by replacing $\phi$ with its approximation $\widetilde{\phi}_\alpha$. Let the resulting function be $\widetilde{g}_q \in \mathcal{ST}^{2,1,3}$.

Then, it is easy to check that $g_q(\boldsymbol{X} + \boldsymbol{E}) = \widetilde{g}_q(\boldsymbol{X} + \boldsymbol{E})$ holds if all coordinates of $\boldsymbol{X} \in [0,1)^{d \times n}$ are in the intervals of the form $[k\delta, (k+1-\alpha)\delta]$ for some $k \in [0 : n/\delta - 1]$; i.e., the intervals in which $\widetilde{\phi}_\alpha$ perfectly approximates $\phi$. The Lebesgue measure of the set of such inputs $\boldsymbol{X}$ is

$$((1-\alpha)\delta)^{nd} \times \frac{1}{\delta^{nd}} = (1-\alpha)^{nd},$$

and this can be made arbitrarily close to 1 by making $\alpha$ small. As a result, "most" of the input $\boldsymbol{X} \in \mathbb{D}$ satisfies $g_q(\boldsymbol{X} + \boldsymbol{E}) = \widetilde{g}_q(\boldsymbol{X} + \boldsymbol{E}) \in \mathbb{H}_\delta$, while a small fraction (of measure at most $1 - (1-\alpha)^{nd}$) can map to some other values. For most of the remaining of the proof, we will consider the fraction of inputs mapped correctly to $\mathbb{H}_\delta$ and bound their approximation error. We will come back to the $1 - (1-\alpha)^{nd}$ fraction at the end of the proof.

### F.2 Approximating the contextual mapping $g_c$ (Lemma 7)

Let us now consider approximating the contextual mapping $g_c$ in Lemma 7, constructed using the hardmax $\sigma_H$ operators, with the standard sparse self-attention layers employing $\rho$ operator. We will call the approximation $\widetilde{g}_c$. Recall that $\rho$ satisfies Assumption 2:

**Assumption 2.** *For any $\zeta > 0$ and $\eta \in (0,1]$, $\exists\, t > 0$ such that, for any column input $\boldsymbol{v}$ satisfying $v_{j^*} - \max_{j \ne j^*} v_j \ge \zeta$ (where $j^* = \arg\max_j v_j$), we have $\rho[t\boldsymbol{v}]_{j^*} \ge 1 - \eta$ and $\sum_{j \ne j^*} \rho[t\boldsymbol{v}]_j \le \eta$.*

This means that $\rho$ can closely approximate $\sigma_H$ in the sense that whenever the input vector $\boldsymbol{v}$ to the $\rho$ operator has a maximum element $v_{j^*}$ by some margin $\zeta$, then the $j^*$-th component of the output $\rho[t\boldsymbol{v}]$ is close to 1, while the other components of $\rho[t\boldsymbol{v}]$ are close to 0.

Recall that $g_c$ consists of two parts. The first part is a composition of sparse selective shift operations, and the second is a composition of all-max-shift operations. We will first examine how "errors" are introduced when $\sigma_H$ is replaced with $\rho$ in both operations, discuss how the errors accumulate, and show how to choose the right $\zeta$ and $\eta$ to control the errors in the approximation $\widetilde{g}_c$.

**Errors introduced by $\rho$: Sparse selective shift operation.** Recall that the key component in both the selective shift operation and all-max-shift operation is the sparse attention head $\psi^l(\cdot)$, which computes its $k$-th column as the following:

$$\psi^l(\boldsymbol{Z}; b_Q)_k := \boldsymbol{u}^T \boldsymbol{Z}_{\mathcal{A}_k^l} \sigma_H[(\boldsymbol{u}^T \boldsymbol{Z}_{\mathcal{A}_k^l})^T (\boldsymbol{u}^T \boldsymbol{Z}_k - b_Q)] = \begin{cases} \max_{j \in \mathcal{A}_k^l} \boldsymbol{u}^T \boldsymbol{Z}_j & \text{if } \boldsymbol{u}^T \boldsymbol{Z}_k > b_Q, \\ \min_{j \in \mathcal{A}_k^l} \boldsymbol{u}^T \boldsymbol{Z}_j & \text{if } \boldsymbol{u}^T \boldsymbol{Z}_k < b_Q. \end{cases}$$

Now suppose we replaced $\sigma_H$ with $\rho$ satisfying Assumption 2. Suppose each entry in $\boldsymbol{u}^T \boldsymbol{Z}$ differs at least by $\delta$, which is true in the construction of $g_c$. We choose $\zeta = \delta/2$ and some $0 < \eta < 1$, and corresponding $t > 0$. Then, replace $\sigma_H[\cdot]$ with $\rho[t\cdot]$ and define

$$\widetilde{\psi}^l(\boldsymbol{Z}; b_Q)_k := \boldsymbol{u}^T \boldsymbol{Z}_{\mathcal{A}_k^l} \rho[t(\boldsymbol{u}^T \boldsymbol{Z}_{\mathcal{A}_k^l})^T (\boldsymbol{u}^T \boldsymbol{Z}_k - b_Q)].$$

If $\boldsymbol{u}^T \boldsymbol{Z}_k > b_Q$, it is easy to check that $\widetilde{\psi}^l(\boldsymbol{Z}; b_Q)_k$ satisfies

$$(1 - \eta) \max_{j \in \mathcal{A}_k^l} \boldsymbol{u}^T \boldsymbol{Z}_j + \eta \min_{j \in \mathcal{A}_k^l} \boldsymbol{u}^T \boldsymbol{Z}_j \le \widetilde{\psi}^l(\boldsymbol{Z}; b_Q)_k \le \max_{j \in \mathcal{A}_k^l} \boldsymbol{u}^T \boldsymbol{Z}_j. \tag{15}$$

Similarly, if $\boldsymbol{u}^T\boldsymbol{Z}_k < b_Q$, we have

$$\min_{j\in\mathcal{A}_k^l}\boldsymbol{u}^T\boldsymbol{Z}_j \le \widetilde{\psi}^l(\boldsymbol{Z};b_Q)_k \le (1-\eta)\min_{j\in\mathcal{A}_k^l}\boldsymbol{u}^T\boldsymbol{Z}_j + \eta\max_{j\in\mathcal{A}_k^l}\boldsymbol{u}^T\boldsymbol{Z}_j.$$

Now consider the *approximate* sparse selective shift operator $\widetilde{\Psi}^l$, implemented with $\widetilde{\psi}^l$. For $b_Q < b_Q'$, we define

$$\widetilde{\Psi}^l(\boldsymbol{Z};c,b_Q,b_Q') := \boldsymbol{Z} + \begin{bmatrix} c\boldsymbol{e}^{(1)} & -c\boldsymbol{e}^{(1)} \end{bmatrix}\begin{bmatrix} \widetilde{\psi}^l(\boldsymbol{Z};b_Q) \\ \widetilde{\psi}^l(\boldsymbol{Z};b_Q') \end{bmatrix}.$$

For any column $\boldsymbol{Z}_k$ satisfying $b_Q < \boldsymbol{u}^T\boldsymbol{Z}_k < b_Q'$, we have

$$(1-2\eta)\left(\max_{j\in\mathcal{A}_k^l}\boldsymbol{u}^T\boldsymbol{Z}_j - \min_{j\in\mathcal{A}_k^l}\boldsymbol{u}^T\boldsymbol{Z}_j\right) \le \widetilde{\psi}^l(\boldsymbol{Z};b_Q)_k - \widetilde{\psi}^l(\boldsymbol{Z};b_Q')_k \le \max_{j\in\mathcal{A}_k^l}\boldsymbol{u}^T\boldsymbol{Z}_j - \min_{j\in\mathcal{A}_k^l}\boldsymbol{u}^T\boldsymbol{Z}_j,$$

and for any column $\boldsymbol{Z}_k$ satisfying $\boldsymbol{u}^T\boldsymbol{Z}_k \notin [b_Q,b_Q']$, we get

$$|\widetilde{\psi}^l(\boldsymbol{Z};b_Q)_k - \widetilde{\psi}^l(\boldsymbol{Z};b_Q')_k| \le \eta\left(\max_{j\in\mathcal{A}_k^l}\boldsymbol{u}^T\boldsymbol{Z}_j - \min_{j\in\mathcal{A}_k^l}\boldsymbol{u}^T\boldsymbol{Z}_j\right).$$

Recall that for the hardmax $\sigma_{\mathrm{H}}$ version, we had

$$\psi^l(\boldsymbol{Z};b_Q)_k - \psi^l(\boldsymbol{Z};b_Q')_k = \begin{cases} \max_{j\in\mathcal{A}_k^l}\boldsymbol{u}^T\boldsymbol{Z}_j - \min_{j\in\mathcal{A}_k^l}\boldsymbol{u}^T\boldsymbol{Z}_j & \text{if } b_Q < \boldsymbol{u}^T\boldsymbol{Z}_k < b_Q', \\ 0 & \text{if } \boldsymbol{u}^T\boldsymbol{Z}_k \notin [b_Q,b_Q']. \end{cases}$$

From this observation, the approximation error $\widetilde{\Psi}^l - \Psi^l$ of the selective shift operator on the $(j,k)$-th entry of the output can be bounded as follows:

$$\widetilde{\Psi}^l(\boldsymbol{Z};c,b_Q,b_Q')_{j,k} - \Psi^l(\boldsymbol{Z};c,b_Q,b_Q')_{j,k} \in \begin{cases} [-2c\eta D_k^l, 0] & \text{if } j=1, \boldsymbol{u}^T\boldsymbol{Z}_k \in (b_Q,b_Q'), \\ [-c\eta D_k^l, c\eta D_k^l] & \text{if } j=1, \boldsymbol{u}^T\boldsymbol{Z}_k \notin [b_Q,b_Q'], \\ \{0\} & \text{if } j\neq 1, \end{cases}$$

where we used $D_k^l := \max_{j\in\mathcal{A}_k^l}\boldsymbol{u}^T\boldsymbol{Z}_j - \min_{j\in\mathcal{A}_k^l}\boldsymbol{u}^T\boldsymbol{Z}_j$ for simplicity.

**Errors introduced by $\rho$: All-max-shift operation.** Next, we examine the approximation error of the all-max-shift operation introduced by replacement of $\sigma_{\mathrm{H}}$ with $\rho$. Let us define the *approximate* all-max-shift operation $\widetilde{\Omega}^l$:

$$\widetilde{\Omega}^l(\boldsymbol{Z};c) = \boldsymbol{Z} + c\boldsymbol{e}^{(1)}\widetilde{\psi}^l(\boldsymbol{Z};0).$$

From (15), we can check that the approximation error $\widetilde{\Omega}^l - \Omega^l$ of the all-max-shift operation is bounded as

$$\widetilde{\Omega}^l(\boldsymbol{Z};c)_{j,k} - \Omega^l(\boldsymbol{Z};c)_{j,k} \in \begin{cases} [-c\eta D_k^l, 0] & \text{if } j=1, \\ \{0\} & \text{if } j\neq 1. \end{cases}$$

**Errors in selective shift operations.** Given these approximation error bounds of single operations, we now analyze the accumulation of errors through multiple layers. We first consider the first $p\delta^{-d}$ self-attention layers in $g_c$. Recall that they consist of selective shift layers $\Psi^{l_2}(\cdot;\delta^{-d},b-\delta/2,b+\delta/2)$ for $b\in[0:\delta:\delta^{-d+1}-\delta]$ and $(p-1)\delta^{-d}$ identity layers. A natural way to approximate these layers with standard self-attention layers is to use approximate layers $\widetilde{\Psi}^{l_2}(\cdot;\delta^{-d},b-\delta/2,b+\delta/2)$, with sufficiently large $t > 0$. As we have seen above, there is no error introduced by $\rho$ except for the first row. Thus, we will analyze the approximation error of $\widetilde{\Psi}^{l_2}(\cdot;\delta^{-d},b-\delta/2,b+\delta/2)$ for the first row only.

Let us remind the readers how the first selective shift operation (done by the first $p\delta^{-d}$ layers) originally worked in $g_c$. The input to $g_c$ is $\boldsymbol{H}$, and we define $z_k := \boldsymbol{u}^T\boldsymbol{H}_k$ and $\Delta = \sum_{i=0}^{d-1}\delta^{-i}$. Recall from Eqs. (7) and (8) in § E.2 that

$$0 \le z_{\gamma(2)} < z_{\gamma(3)} < \cdots < z_{\gamma(n)} < z_{\gamma(1)} \le (n-1)\Delta + \delta^{-d+1} - \delta < n\delta^{-d}$$

and $z_{\gamma(2)} \in [0 : \delta : \delta^{-d+1} - \delta]$, so $z_{\gamma(2)}$ will undergo the selective shift by one of the self-attention layers, which updates the $(1, \gamma(2))$-th entry of the input. Let $\widetilde{\boldsymbol{H}}_{\gamma(2)}$ be the updated value of the column and $\widetilde{z}_{\gamma(2)} := \boldsymbol{u}^T \widetilde{\boldsymbol{H}}_{\gamma(2)}$. The new sequence satisfies

$$\Delta \leq z_{\gamma(3)} < \cdots < z_{\gamma(n)} < z_{\gamma(1)} < \widetilde{z}_{\gamma(2)} < n\delta^{-2d},$$

where the strict upper bound on $\widetilde{z}_{\gamma(2)}$ is from Eq. (11).

In case of the approximation $\widetilde{\Psi}^{l_2}$, we have seen that the error depends on the gap between maximum and minimum of $\boldsymbol{u}^T \boldsymbol{Z}_j$'s, and this gap may grow larger as error accumulates; in the worst case, it may grow exponentially. To see this, suppose $a_0$ and $b_0$ are the maximum and minimum value of $\boldsymbol{u}^T \boldsymbol{Z}_j$'s, and they go through a selective shift operation, but they do not belong to the range of the operation $(b_Q, b'_Q)$. Then, $a_0$ and $b_0$ will be updated to $a_1$ and $b_1$, which are bounded by

$$a_1 \leq a_0 + \delta^{-d}\eta(a_0 - b_0), \quad b_1 \geq b_0 - \delta^{-d}\eta(a_0 - b_0).$$

After the next layer, we get

$$a_2 \leq a_1 + \delta^{-d}\eta(a_1 - b_1) \leq a_0 + \delta^{-d}\eta(a_0 - b_0) + \delta^{-d}\eta(1 + 2\delta^{-d}\eta)(a_0 - b_0),$$
$$b_2 \geq b_1 - \delta^{-d}\eta(a_1 - b_1) \geq b_0 - \delta^{-d}\eta(a_0 - b_0) - \delta^{-d}\eta(1 + 2\delta^{-d}\eta)(a_0 - b_0).$$

Similarly, after $k$ such layers, we get

$$a_k \leq a_0 + (a_0 - b_0)\delta^{-d}\eta \sum_{i=0}^{k-1}(1 + 2\delta^{-d}\eta)^i,$$

$$b_k \geq b_0 - (a_0 - b_0)\delta^{-d}\eta \sum_{i=0}^{k-1}(1 + 2\delta^{-d}\eta)^i,$$

showing that the gap $a_k - b_k$ may grow exponentially in the worst case:

$$a_k - b_k \leq (1 + 2\delta^{-d}\eta)^k(a_0 - b_0).$$

In the error-less case ($\sigma_{\mathrm{H}}$), for any input sequence $\boldsymbol{H}$, the maximum possible difference between maximum and minimum of $\boldsymbol{u}^T \boldsymbol{H}$ is bounded above by $n\delta^{-d}$, and after one selective shift operation was done on the $\gamma(2)$-th column, the difference is then bounded by $n\delta^{-2d}$. Therefore, the worst-case possible error introduced by $\rho$ is bounded above by the sum of the worst-case errors calculated assuming that we started off with max-min difference $n\delta^{-2d}$. Using this observation, the error on each first-row entry of the sequence after the first $p\delta^{-d}$ layers is bounded above by

$$2n\delta^{-2d} \cdot \delta^{-d}\eta \sum_{i=0}^{\delta^{-d}-1}(1 + 2\delta^{-d}\eta)^i, \tag{16}$$

where a factor of 2 is introduced because when the selective shift operation is applied to the $\gamma(2)$-th column, it may introduce an error which is twice the magnitude of the error introduced to the other columns. We want to make (16) smaller than $\frac{\delta}{8n}$. By Assumption 2, we can always choose $t > 0$ that satisfies the assumption for

$$\zeta = \frac{\delta}{2}, \text{ and } \eta = \tfrac{1}{2}\delta^{2d} \log\left(1 + \frac{\delta^{2d}\widetilde{\delta}}{8n^2}\right) > 0, \text{ where } \widetilde{\delta} := \min\left\{\delta, \frac{2^{1-1/p}\epsilon}{n^{1/p}}\right\}.$$

Using such $t$, we can control the total accumulated error by the first $p\delta^{-d}$ selective shift operations below $\frac{\widetilde{\delta}}{8n}$:

$$2n\delta^{-2d} \cdot \delta^{-d}\eta \sum_{i=0}^{\delta^{-d}-1}(1 + 2\delta^{-d}\eta)^i \leq 2n\delta^{-3d}\eta \frac{(1 + 2\delta^{-d}\eta)^{\delta^{-d}} - 1}{(1 + 2\delta^{-d}\eta) - 1}$$

$$= n\delta^{-2d}\left(\left(1 + \frac{\log\left(1 + \frac{\delta^{2d}\widetilde{\delta}}{8n^2}\right)}{\delta^{-d}}\right)^{\delta^{-d}} - 1\right) \leq n\delta^{-2d}\left(\exp\log\left(1 + \frac{\delta^{2d}\widetilde{\delta}}{8n^2}\right) - 1\right)$$

$$= n\delta^{-2d}\frac{\delta^{2d}\widetilde{\delta}}{8n^2} = \frac{\widetilde{\delta}}{8n}.$$

Therefore, after the first $p\delta^{-d}$ selective shift layers, the accumulated error for each entry of the first row is at most $\widetilde{\delta}/8n$.

We can also apply similar arguments to the remaining selective shift layers. For example, for the $j$-th set of $p\delta^{-d}$ selective shift layers where the operation is done on $\gamma(j+1)$-th column of the input, the gap between the maximum and the minimum, including the accumulated error from previous layers, is bounded above by $n\delta^{-(j+1)d}$. Therefore, for this set of layers, the maximum accumulated error is bounded by

$$2n\delta^{-(j+1)d} \cdot \delta^{-d}\eta \sum_{i=0}^{\delta^{-d}-1}(1+2\delta^{-d}\eta)^i.$$

So, choosing $t > 0$ that satisfies Assumption 2 for $\eta = \frac{\delta}{2}$ and $\eta = \frac{1}{2}\delta^{2d}\log(1+\frac{\delta^{(j+1)d}\widetilde{\delta}}{8n^2})$, we can control the accumulated error introduced by the $p\delta^{-d}$ layers below $\frac{\widetilde{\delta}}{8n}$:

$$2n\delta^{-(j+1)d} \cdot \delta^{-d}\eta \sum_{i=0}^{\delta^{-d}-1}(1+2\delta^{-d}\eta)^i \le 2n\delta^{-(j+2)d}\eta\frac{(1+2\delta^{-d}\eta)^{\delta^{-d}}-1}{(1+2\delta^{-d}\eta)-1}$$

$$\le n\delta^{-(j+1)d}\left(\left(1+\frac{\log\left(1+\frac{\delta^{(j+1)d}\widetilde{\delta}}{8n^2}\right)}{\delta^{-d}}\right)^{\delta^{-d}}-1\right) \le \frac{\widetilde{\delta}}{8n}.$$

In total, the accumulated error by the first $p(n-1)/\delta^d$ layers, which correspond to the selective shift operation part of the construction, is at most $\frac{(n-1)\widetilde{\delta}}{8n} \le \frac{\widetilde{\delta}}{8}$.

**Errors in all-max-shift operations.** For all-max-shift operations, we approximate the hardmax $\sigma_H$ all-max-shift operations $\Omega^l(Z; n\delta^{-nd})$ with its $\rho$-counterparts, $\widetilde{\Omega}^l(Z; n\delta^{-nd})$. We can similarly bound the accumulated error in the all-max-shift operations. Recall from § E.2 that during the whole series of all-max-shift operations, the maximum entry in the sequence is upper-bounded by $(2sn\delta^{-nd-1})^s n\delta^{-nd}$ and minimum entry is lower-bounded by $(n-1)\Delta$. Therefore, the gap between the max and min elements, taking into consideration the errors from selective shift operations, is bounded from above by $(2sn\delta^{-nd-1})^s n\delta^{-nd}$. Then, using a similar argument as the select shift operation layers, the maximum error is bounded above by

$$(2sn\delta^{-nd-1})^s n\delta^{-nd} \cdot n\delta^{-nd}\eta \sum_{i=0}^{s-1}(1+n\delta^{-nd}\eta)^i,$$

and we want to make it smaller than $\frac{\widetilde{\delta}}{8}$. By Assumption 2, we can always choose $t > 0$ that satisfies the assumption for

$$\zeta = \frac{\delta}{2}, \text{ and } \eta = \frac{\delta^{nd}}{sn}\log\left(1+\frac{\delta^{s(nd+1)+nd}\widetilde{\delta}}{2^{s+3}s^s n^{s+1}}\right) > 0.$$

Using such $t$, we can control the total accumulated error by the first $p\delta^{-d}$ selective shift operations below $\frac{\widetilde{\delta}}{8}$:

$$(2sn\delta^{-nd-1})^s n\delta^{-nd} \cdot n\delta^{-nd}\eta \sum_{i=0}^{s-1}(1+n\delta^{-nd}\eta)^i$$

$$\le (2sn\delta^{-nd-1})^s n\delta^{-nd} \cdot n\delta^{-nd}\eta\frac{(1+n\delta^{-nd}\eta)^s-1}{(1+n\delta^{-nd}\eta)-1}$$

$$= (2sn\delta^{-nd-1})^s n\delta^{-nd}\left(\left(1+\frac{\log\left(1+\frac{\delta^{s(nd+1)+nd}\widetilde{\delta}}{2^{s+3}s^s n^{s+1}}\right)}{s}\right)^s-1\right)$$

$$\le (2sn\delta^{-nd-1})^s n\delta^{-nd}\frac{\delta^{s(nd+1)+nd}\widetilde{\delta}}{2^{s+3}s^s n^{s+1}} = \frac{\widetilde{\delta}}{8}.$$

So far, we have analyzed the total accumulated error of approximating the contextual mapping function $g_c$ (constructed with hardmax $\sigma_H$) with an approximation $\widetilde{g}_c$ (constructed with $\rho$). We have seen that for any input $\boldsymbol{H} \in \mathbb{H}_\delta$, the approximation error can be controlled so that the error by the selective shift operation part is at most $\widetilde{\delta}/8$ and the all-max-shift operation part is at most $\widetilde{\delta}/8$. Therefore, the total error of the $(j, k)$-th entry can be bounded as

$$\widetilde{g}_c(\boldsymbol{H})_{j,k} - g_c(\boldsymbol{H})_{j,k} \in \begin{cases} [-\frac{\widetilde{\delta}}{4}, \frac{\widetilde{\delta}}{4}] & j = 1, \\ \{0\} & j \neq 1, \end{cases}$$

for any $\boldsymbol{H} \in \mathbb{H}_\delta$.

### F.3  Approximating the value mapping $g_v$ (Lemma 8)

We now consider the approximation of the value mapping $g_v$ with standard feed-forward layers. In $g_v$, we implemented the function with layers of the form

$$\boldsymbol{Z} \mapsto \boldsymbol{Z} + (\overline{f}(\boldsymbol{H} - \boldsymbol{E})_k - g_c(\boldsymbol{H})_k)\phi'(\boldsymbol{u}^T \boldsymbol{Z} - \boldsymbol{u}^T g_c(\boldsymbol{H})_k \mathbf{1}_n^T), \quad \phi'(t) = \begin{cases} 0 & t < -\delta/2 \text{ or } t \geq \delta/2, \\ 1 & -\delta/2 \leq t < \delta/2. \end{cases}$$

Since the output of contextual mapping $g_c(\boldsymbol{H})$ and its approximation $\widetilde{g}_c(\boldsymbol{H})$ differ in only the first row and by $\widetilde{\delta}/4 \leq \delta/4$, one can approximate each layer in $g_v$ by replacing $\phi'$ with an approximation $\widetilde{\phi}'$, implementable with four ReLU's:

$$\widetilde{\phi}'(t) = \begin{cases} 0 & t < -\delta/2 \text{ or } t \geq \delta/2, \\ \frac{4}{\delta}t + 2 & -\delta/2 \leq t < -\delta/4, \\ 1 & -\delta/4 \leq t < \delta/4, \\ -\frac{4}{\delta}t + 2 & \delta/4 \leq t < \delta/2. \end{cases}$$

Let $\widetilde{g}_v$ be the approximation of $g_v$ constructed this way. Because the error on $\widetilde{g}_c$ is bounded by $\widetilde{\delta}/4$, the error on the final output $\widetilde{g}_v$ is also bounded by $\widetilde{\delta}/4$. That is, for any $\boldsymbol{H} \in \mathbb{H}_\delta$,

$$\widetilde{g}_v(\widetilde{g}_c(\boldsymbol{H}))_{j,k} - g_v(g_c(\boldsymbol{H}))_{j,k} \in \begin{cases} [-\frac{\widetilde{\delta}}{4}, \frac{\widetilde{\delta}}{4}] & j = 1, \\ \{0\} & j \neq 1. \end{cases}$$

Hence, using $\widetilde{\delta} := \min\left\{\delta, \frac{2^{1-1/p}\epsilon}{n^{1/p}}\right\}$, we have

$$\|\widetilde{g}_v(\widetilde{g}_c(\boldsymbol{H})) - g_v(g_c(\boldsymbol{H}))\|_p^p \leq n\left(\frac{\widetilde{\delta}}{4}\right)^p \leq \frac{1}{2}\left(\frac{\epsilon}{2}\right)^p,$$

for all $\boldsymbol{H} \in \mathbb{H}_\delta$.

### F.4  Finishing the proof

Recall from § F.1 that the approximated quantization function $\widetilde{g}_q$ maps most of the input $\boldsymbol{X} \in \mathbb{D}$ to $\boldsymbol{H} \in \mathbb{H}_\delta$, and a small fraction of them (of measure at most $1 - (1-\alpha)^{nd}$) to something else. Note now that the original function $\overline{g} = g_v \circ g_c \circ g_q$ and the approximation $g = \widetilde{g}_v \circ \widetilde{g}_c \circ \widetilde{g}_q$ are both bounded, so there is a global constant $B$ such chat $\|\overline{g}(\boldsymbol{X} + \boldsymbol{E}) - g(\boldsymbol{X} + \boldsymbol{E})\|_p \leq B$ for all $\boldsymbol{X} \in \mathbb{D}$.

We can divide the integral over $\mathbb{D}$ to two disjoint sets. The first one $\mathbb{D}_1 := \{\boldsymbol{X} \in \mathbb{D} \mid \widetilde{g}_q(\boldsymbol{X} + \boldsymbol{E}) \in \mathbb{H}_\delta\}$ is the set of input $\boldsymbol{X}$ mapped to $\mathbb{H}_\delta$ by $\widetilde{g}_q$, and the other is its complement $\mathbb{D}_2 = \mathbb{D} \setminus \mathbb{D}_1$.

$$\begin{aligned} \mathsf{d}_p(\overline{g}, g)^p &:= \int_{\mathbb{D}} \|\overline{g}(\boldsymbol{X} + \boldsymbol{E}) - g(\boldsymbol{X} + \boldsymbol{E})\|_p^p \, d\boldsymbol{X} \\ &= \int_{\mathbb{D}_1} \|\overline{g}(\boldsymbol{X} + \boldsymbol{E}) - g(\boldsymbol{X} + \boldsymbol{E})\|_p^p \, d\boldsymbol{X} + \int_{\mathbb{D}_2} \|\overline{g}(\boldsymbol{X} + \boldsymbol{E}) - g(\boldsymbol{X} + \boldsymbol{E})\|_p^p \, d\boldsymbol{X} \\ &\leq \frac{1}{2}\left(\frac{\epsilon}{2}\right)^p + (1 - (1-\alpha)^{nd})B^p. \end{aligned}$$

One can make $\alpha$ close enough to 1 so that the second term is less than $\frac{1}{2}\left(\frac{\epsilon}{2}\right)^p$. This makes $\mathsf{d}_p(\overline{g}, g) \leq \epsilon/2$, hence finishing the proof.

# G Experimental setup

## G.1 Copying task

We generated the synthetic dataset for the copying task. The input sequence to the copying task has the format 0s0s, where s is a 127 length sequence of symbols randomly sampled from the range of $[0, 127]$. The training set contains 100K sequences, while the testing set contains 10K sequences.

We implement the copying task as a masked-LM [10] style prediction task by masking all the tokens in the second half of the sequence. For the test examples, each masked token is predicted independently. For the results reported in § 5, we experiment with bidirectional models, where each token can attend to both previous and future tokens.

The maximum sequence length is $n = 256$, and we use embedding dimension $d = 256$. The model has 1 to 4 attention layers with $h = 4$ attention heads of size $m = 64$, followed by a feed-forward hidden layer of size $r = 512$. We train the model with the AdamW optimizer with weight decay and no dropout. We train the model using 3,000 warmup steps and a total of 500K training steps. The learning rate is $1e^{-4}$. We use the batch size 1,024 on 8 TPUv3 chips.

For all sparsity patterns other than the RANDOM pattern, we choose the segment length $w$ to be 16 for all patterns. This segment length results in the sparsest level for the STRIDED and FIXED patterns. In Table 1, we include the sparsity level as a reference. For this task, we report the prediction accuracy for all the tokens.

## G.2 Language modeling

For the language modeling task, we train on the One Billion Word Benchmark [5] which contains almost one billion tokens and a vocabulary of more than 800K tokens.

We use the Transformer model in the Tensor2Tensor framework [29]. We use a 12-block (cf. (2)) Transformer, with embedding dimension $d = 256$, maximum sequence length $n = 256$, number of heads $h = 8$, head size $m = 64$, and feed-forward hidden layer size $r = 1024$. Since language modeling task is auto-regressive (attending to only past tokens) in nature, we evaluate the (sparse) attention score matrices and mask them to be an upper-triangular matrix. We train the model with the Adafactor with weight decay. We train the model using 10K warmup steps and a total of 240K steps. We use the batch size 4,096 on 8 TPUv2 chips.

For this task, we report the perplexity.

## G.3 Translation

For the translation task, we train on the WMT18 en-cs datasets (Europarl v7, Common Crawl corpus, News Commentary v13, and CzEng), with a total of 15M pairs of sentences, and test on the newstest2015 en-cs dataset, with 2,656 pairs.

We use the encoder-decoder architecture and apply the sparse attention on both encoder and decoder. We use the Transformer model in the Tensor2Tensor framework [29] and the same setup as the language modeling task, except for having 6 blocks in the Transformer networks, with head size $m = 32$ and having autoregressive patterns only in decoders.

For this task, we report the cased BLEU score.

## G.4 GLUE tasks

For the GLUE tasks, we use the pre-training and fine-tuning framework [10]. Following Devlin et al. [10] we first pre-train a BERT$_{BASE}$ model for 450K steps on the BooksCorpus [36] (800M words) and the English Wikipedia datasets (2,500M words). We later finetune the model on data from each task separately. For each setting, we use the same sparsity pattern and head configuration in both the pre-training and the fine-tuning stages. The sequence length is $n = 128$ in both stages.

We report the average accuracy of three runs on the dev set for all tasks. For each setting, we pre-train a model and run fine-tuning three times.

**Table 2.** Accuracy on the synthetic copying task when using an auto-regressive model. Percentages in parentheses mark the sparsity levels.

| | STRIDED | | | FIXED | | | STAR | RANDOM |
| **Depth** | UNION (87%) | MULTIHEAD (93%) | SEQUENTIAL (93%) | UNION (87%) | MULTIHEAD (93%) | SEQUENTIAL (93%) | (87%) | (90%) |
|---|---|---|---|---|---|---|---|---|
| 1-layer | 0.79% | 0.78% | 0.78% | 7.02% | 7.04% | 0.81% | 0.77% | 33.13% |
| 2-layer | 12.40% | 8.26% | 1.57% | 73.43% | 13.24% | 92.10% | 12.32% | 67.30% |
| 3-layer | 94.50% | 65.58% | 60.88% | 99.87% | 70.82% | 99.84% | 14.03% | 89.50% |
| 4-layer | 100% | 100% | 98.40% | 99.97% | 99.16% | 99.97% | 31.19% | 95.88% |

(a) WMT en-de      (b) WMT de-en

**Figure 3.** Comparison of sparsity patterns and different head configurations on the WMT de-en and en-de translation tasks.

(a) CoLA      (b) MRPC

**Figure 4.** Comparison of sparsity patterns and different head configurations on the CoLA and MRPC tasks for the BERT$_{\text{BASE}}$ model.

# H  Additional experimental results

We report additional experimental results in this section.

## H.1  Copying task

We include the results for the copying task using auto-regressive (unidirectional) models as in LM, where each token can only attend to previous tokens, in Table 2. In this case, the STAR pattern cannot attend to the last replay token. Indeed, the STAR pattern shows better performance when the model is bidirectional (cf. Table 1).

## H.2  Translation

We present experimental results of the translation tasks on the WMT English-German and German-English datasets in Figure 3. We train on WMT18 (Europarl v7, Common Crawl corpus and News Commentary v13) and test on newstest 2015 datasets. The figures show similar trends to the results on the WMT en-cs dataset in Figure 1b.

## H.3 GLUE tasks

Figure 4 presents the results comparing the sparsity patterns and the head configurations on the CoLA and MRPC tasks using the BERT$_{\text{BASE}}$ model. CoLA is a single-sentence classification task, asking if a sentence is a grammatical English sentence. MRPC is a sentence-pair classification task, where each example is a pair of sentences and the label indicates whether the sentences are semantically equivalent.