[Reviews · NeurIPS 2020]

Review 1

Summary and Contributions: The authors prove that sparse transformers (transformers in which the potential connections in the attention layer are limited) can 'universally approximate' functions over sequences of finite length - under certain asusmptions over which connections are allowed to remain. Previously, universal approximability has only been shown for 'fully connected' transformers. They present some experiments demonstrating the strength of sparse transformers on various tasks, comparing between different sparsity patterns and to dense transformers.

Strengths: The claim that even (certain) sparse transformers may still universally approximate sequence-to-sequence functions (over sequences of finite length) is non-trivial. Indeed without it, one might intuitively guess that sparsity would reduce the expressive power of a transformer, and see this as an argument against using them. As sparse transformers have less connections, they require less computations to use than a 'regular' transformer, (all other hyperparameters of the networks being equal), making them prefereable in that sense. Hence it is valuable to see that they can be used without concern of losing power.

Weaknesses: 1. Reading this paper, you are inclined to believe that (certain) sparse transformers can be both smaller than (due to their sparsity) and as powerful as (due to the result of this work) 'standard' transformers. However, on second look I have noticed that the claims and constructed transformers, at least as presented in the main body of the work, do not discuss the number of layers that the transformers need in order to achieve this result. This makes it harder to make meaningful conclusions about sparse transformers: what if transformers can be as powerful when the number of connections in the attention has been reduced from O(n^2) (full) to O(n), but this requires a factor of O(n) more layers? A clear discussion of the number of layers used in the constructions, in the main body of the paper, will be really valuable in order to understand whether we should be 'transitioning' to sparse transformers or not. (A result in either direction would be meaningful, and not detracting from the result itself.) 2. The proof of the claim is mostly presented in the (very long) appendix. Reading the paper alone, unfortunately, is not enough to fully convey the method. However, I am not sure if this can really be remedied given the page limit.

Correctness: As so much is left in the appendix, I was unable to truly check the correctness of this paper. With respect to the parts presented in the main section: the formal framework presented for discussion of sparse transformers is neat and well constructed, the assumptions (with one comment below) under which the claim works are clearly presented and generally reasonable, and the main claim and proof sketch are well presented and plausible.

Clarity: The writing in the main paper is generally clear, although it would benefit from expansions/intuitions/corrections as follows: 1. An intuitive discussion of the purpose of the assumptions, especially assumption (1.2) (the meaning of the assumptions is quite clear, but the reason for their necessity is not). 2. A discussion, presented in the main body of the paper, of how many layers are used in the constructions for the proofs. 3. In step 2, ReLU is replaced with piecewise functions that have 3 pieces, despite ReLU itself having only 2 pieces. Some discussion explaining the intuition behind this would be very useful. 4. If there is any additional space, I think a general expansion on what is happening in step 2 in the proof, and of paragraph 259-267, would be very helpful. Additionally, a couple of sentences are unclear: 1. Line 297 - 'one sparse transformer block' - is the meaning that only one of the layers in the transformer is sparse? Or the transformer has only 1 layer? Or that each transformer has only one sparsity pattern? 2. Line 265 - 'every other token' in the sequence can sound like 'one token yes, one token no, yes, no, etc', though I think the meaning is "all the other tokens", maybe change? Note: I have not checked the clarity of the supplementary material, which the paper heavily relies on.

Relation to Prior Work: This work extends a previous result -- about dense transformers -- to apply also to sparse transformers. The differences and similarities between the two works are quite clear and also explicitly presented by the authors. The work mentions, but does not really compare to another theoretical transformer-expressiveness work, which is a shame: in the paper 'Theoretical limitations of self-attention in neural sequence models', Hahn shows (for instance) that a transformer with hard attention is unable to recognise parity for arbitrary length strings, and that for some tasks the accuracy of a transformer with soft attention will reduce as strings become longer. As a knee jerk reaction, one might think this contradicts with the findings of this work, but in fact it does not: the claims in this work are for approximating functions over sequences of a finite given length, and so any one approximation need only be precise enough for the domain it is over, and is not concerned with increasing input length. I think noting this clearly in the work would be valuable. On a different note, I think that whenever someone works so closely with transformers, and especially attention, it makes sense to cite the original works presenting attention, i.e.: Effective Approaches to Attention-based Neural Machine Translation (Luong et al) Neural Machine Translation by Jointly Learning to Align and Translate (Bahdanau et al)

Reproducibility: Yes

Additional Feedback: In addition to the comments detailed in the above section, I would appreciate the authors sharing any insight they have on the following experimental results: 1. In figure 1b, why does the BLEU score of the strided-multihead transformer increase as the sparsity increases? 2. (This is discussed minimally, but more may be interesting): why does the star-sparsity fail so strongly on the copying task, despite being theoretically as capable as the others? I wonder if this ties in with the discussion of number of layers needed to maintain the universal expressivity once the attention is sparse (mentioned also in 'weaknesses'). Insights the authors may have from the proof construction would be helpful here. (It would be nice if the theory could account for star's relative weakness on this task). ==== I appreciate the clarifications in the author response, I think they should be brought to the main paper too. In particular, the authors should make it clearer how many layers (approximately, if necessary) are needed to construct the approximation, as a function of the various parameters of the construction eg the sequence length n, and other parameters (delta,d,...). Following some discussion, it may also be good to clarify that the universal sequence to sequence approximation is between sequences of same length (i.e. that the discussion is on the encoder).


Review 2

Summary and Contributions: Raised my score after reading the rebuttal. ============== This paper provides a theoretical foundation for sparse transformers: a sparse transformer with O(n) attention connections in each layer can universally approximate any seq-to-seq model. The author extends the previous theory of universal approximation theory of dense transformers to the sparse transformers. In particular, they study the case that sparse attention satisfies several conditions.

Strengths: This paper shows several conditions that sparse transformers needs in order to have the same expressive power as the dense transformers. These conditions may further help future researches on designing sparse transformers. They also empirically compare the performance of the different sparse attention patterns.

Weaknesses: It is a straight-forward extension of [28]. The result is not superising, since the model is allowed to have aritrary depth. The connection between any 2 positions can happen in deep layers. Since the number of layers in sparse transformers are usually very small (10~20) compared with the length of the setences (hundreds of tokens), why sparse transformers still perform similarly to the dense transformer is unclear. It would be more interesting to investigate the model expressive power when the depth is limited. Or how many layers does the transformer need to have the universal approximation ability. If these strong conditions are required for a sparse transfomer to have the same approximation ability as the dense transformer, it would be better to theoretically prove that these conditions are required or empirically show that these conditions can not be broken. Although in real data, the sparse model may work without satifying these conditions. Some experiments with hand-crafted data can also help with showing that these conditions are necessary. Otherwise, I just feel that these assumptions/conditions are just too strong and make the results less interesting. Can the analysis also extend to encoder-decoder attention?

Correctness: Yes.

Clarity: Yes.

Relation to Prior Work: Yes.

Reproducibility: Yes

Additional Feedback:


Review 3

Summary and Contributions: The work claims to have generalized the approximation power of Transformer to the case of sparse attention patterns. These patterns include some of the existing sparse Transformers.

Strengths: The claimed result is new and important to the machine learning community.

Weaknesses: The result is clear and important, but the hardness of the result and the proof remains further explanation. The main point of the proof of the main result (Theorem 1) is to make an additional assumption (Assumption 1.3) on sparse attention which helps the authors to prove Lemma 7 on 'contextual mappings' of a token. As claimed, the key finding of Lemma 7 is that these mappings really reflect the context of a token, and thus different for a token in different sequences/sentences (property 2 in Lemma 7). Moreover, in some direction u, projections of each token are all distinct (property 1 in Lemma 7). Once these contextual mappings are constructed, authors use several position-wise feedforward networks to reach the target function f (Lemma 8). I have the following two concerns on the proof. 1) To my understanding, property 2 in Lemma 7 characterizes a context instead of property 1. However, only property 1 is used in the proof? If so, this could be misleading because the so called "contextual mapping" is not really used in the proof, which makes the notion irrelevant to the result. 2) This problem is more difficult, but will undermine the difficulty of the work. Without loss of generality, in the proof of Lemma 8, let us discuss the case when d=1. In this case property 2 in lemma 7 is equivalent to $g_c(H)_ k$ being all distinct (as a real number), which I denote by $y_k$. The main point of Lemma 8 is to use a stack of position-wise feedforward layers to map y_k to $ \bar{f}(\boldsymbol{H}-\boldsymbol{E})_ {k}$, which I denote by z_k. The problem is, To map $(z_k)_ k$ to $(y_k)_ k$ through a stack of token-wise feedforward layers, why is it necessary to have $z_k$ being distinct for different k?

Correctness: The mathematical claim is correct, but the proof requires more writing and explanation.

Clarity: The results are clearly stated, and the overall idea of the proof is well explained.

Relation to Prior Work: Yes.

Reproducibility: Yes

Additional Feedback:


Review 4

Summary and Contributions: The paper introduces general of sparse attention. Using this framework the authors prove that Transformers with sparse attention are universal approximators under some natural and pretty weak conditions on sparsity patterns.

Strengths: The class of models for which the proof applies includes popular in applications variants of sparse attentions such as Strided and Fixed patterns. The paper shows that it's possible to express any function using sparse attention with O(n) complexity. The interesting part is that one can do so without changing the width of the layers as the input length grows.

Weaknesses: The experiment section is not clearly connected to the theoretical part. For instance, that would be interesting to see whether breaking any of the assumptions of theorem causes significant degradation of the quality, e.g., breaking any clause in assumption 1 for the random connectivity pattern or for some handcrafted ones. The same for assumption 2, e.g., is that true that replacing SM with a hard top-k hard significantly degenerate the performance. Most of the experiments are done for autoregressive case (LM and MT). It seems to break assumption 1.3, as token at first position will never see the whole sequence. If the theory doesn't apply to unidirectional transformers, maybe it's better to switch to naturally bidirectional models such as BERT.

Correctness: The theoretic claims are valid. Empirical methodology is not clear, see above.

Clarity: The paper is well written except for the experiment section. E.g., in G2 it mentioned that masking is used to enforce causality, but there is no mention of masking in G1. So it's not clear how's prediction done in this case. For all masked experiments, it's not clear how STAR attention is adopted. From (6) it follows that every position attends to the last element that is not available in the masked. While a reader can guess that, that'd be great to have more clear explanation. For G3, I assume an encoder-decoder architecture is used. Is sparse attention for both parts or only decoder?

Relation to Prior Work: I believe the paper will also benefit from references to other universal approximators, e.g., RNNs and ResNets. Right now paper only compares with Dense Transformers.

Reproducibility: Yes

Additional Feedback:

[Author Response · NeurIPS 2020]

We would like to thank the reviewers for their valuable comments. Below, we address the concerns raised by the
reviewers. We will make sure to reflect the *comments, suggested references, and our answers* to the final version.

**Response to Reviewer 1.** *Q1-1. Number of layers required?* In § D of the supplementary material, Lemmas 6–8
show that we need $\frac{dn}{\delta}$ sparse Transformer blocks (eq (2)) for quantization, $\frac{p(n-1)}{\delta^d} + s$ for contextual mapping, and
$\frac{n}{\delta^{dn}}$ for value mapping. Recall that $p$ and $s$ are from Assumption 1, and $\delta$ is from Step 1 of § 4.1. In comparison,
§ C of [28] shows that the dense counterpart requires $\frac{dn}{\delta}$, $\frac{n}{\delta^d} + 1$, and $\frac{n}{\delta^{dn}}$ Transformer blocks (eq (1)) for the three
corresponding lemmas. Note two observations: *1) The value mapping **dominates** the depth, and its depth requirements
are **identical** for the two cases*; and *2) For contextual mappings (where the attention layers are used), we need roughly
**p times more** layers for sparse models.* Since $p$ is usually a small constant, these observations mean that sparse
Transformers can achieve universal approximation using depth of the **same order** in $d$ and $n$ as the dense Transformers.

*Q1-2. Re: comments on clarity:* Thank you for the suggestions. In our revision, we plan to add more details of the
assumptions/proofs/references, especially Step 2 of § 4.1. We believe that this will also help remedy Weakness 2 and
better motivate Assumption 1.2. Please also see Q2-2 on the necessity of assumptions. In Line 297, we use 1–4 sparse
attention layers and one feed-forward layer (please see § G.1). In Line 265, we meant "all the other tokens."

**Response to Reviewer 2.** *Q2-1. The result is not surprising, given arbitrary depth:* Our result is *not* a straightfor-
ward extension of [28], as we illustrate in § 4.2; we overcome nontrivial challenges posed by sparsity, which is also
appreciated by Rev1. Although a "connection" between any two positions can happen through multiple sparse layers,
this is only an intuition, and turning it into a rigorous analysis is not easy. Moreover, there are results showing that
*limited width* can render universal approximation *impossible*, **even with arbitrary depth**: see e.g., "Deep, skinny
neural networks are not universal approximators." We would like to emphasize that our paper is the first to provide a
concrete theory justifying sparse attention; our careful analysis reduces the connections per layer from $n^2$ to $O(n)$, with
only $p$ times more attention layers (see Q1-1). Our analysis also gives insights into the design of the sparsity patterns.

*Q2-2. Necessity of assumptions?* We believe that the assumptions are quite reasonable, as also mentioned by Rev1 and
Rev4. Our assumptions are weak enough to be satisfied by many existing sparsity patterns, as we cover in § 3.4. In fact,
we can show that Assumptions 1.1 and 1.3 are **necessary** for universal approximation to hold. For A1.3, assume $n = 2$
and consider a sparsity pattern with $p = 1$: $\mathcal{A}_1^1 = \{1, 2\}, \mathcal{A}_2^1 = \{2\}$. Note that it satisfies A1.1 and A1.2, but not A1.3.
Since the second token never attends to the first token, this sparse Transformer can never approximate a function whose
second output token is dependent on both input tokens; this proves the necessity of A1.3. Next, consider $n = 2$ and
a pattern with $p = 2$: $\mathcal{A}_1^1 = \{1, 2\}, \mathcal{A}_2^1 = \{1, 2\}, \mathcal{A}_1^1 = \{1\}, \mathcal{A}_2^1 = \{1\}$. One can check that this pattern satisfies all
assumptions but A1.1. Since both tokens in the second layer attend to the same single token ($\mathcal{A}_1^2 = \mathcal{A}_2^2 = \{1\}$), the two
tokens in the sequence become identical afterwards, and hence cannot approximate arbitrary functions. As per Rev2's
suggestions on empirical verification, we will add experiments to further validate our assumptions; please also see Q4-1.

*Q2-3. Extension to the encoder-decoder attention?* For now, our analysis applies to the encoder (i.e. BERT-style) part
of the model. Extending this analysis to the encoder-decoder attention would be a very interesting future direction.

**Response to Reviewer 3.** Thanks for the questions. For the hardness of the proof, please refer to Q2-1. Below, we
address the concerns raised; we hope that our answers will clarify the proof and Rev3 will reassess our paper.

*Q3-1. Concern 1)* Both properties are used. § E.3 uses the fact that there are $n|\mathbb{H}_\delta| = n\delta^{-dn}$ distinct real numbers that
$q(\boldsymbol{H})_k := \boldsymbol{u}^T g_c(\boldsymbol{H})_k$ takes (for positions $k \in [n]$ and contexts $\boldsymbol{H} \in \mathbb{H}_\delta$), which is implied by *both* Properties 7.1 & 7.2.

*Q3-2. Concern 2)* In case of $d = 1$, Lemma 8 uses Properties 7.1 & 7.2 to construct a function $g_v$ that maps all $n\delta^{-n}$
distinct values of $y_k := g_c(\boldsymbol{H})_k \in \mathbb{R}$ to $z_k := \overline{f}(\boldsymbol{H} - \boldsymbol{E})_k \in \mathbb{R}$. The reason why $y_k$ must be distinct for different $k$'s
is that the feed-forward layers operate in a *position-wise* manner, hence so does $g_v$; in other words, the same map $g_v^{\text{tkn}}$
is applied to each token. For example, if $y_1 = y_2 = c$, then the first two tokens of the output $g_v(\boldsymbol{y})$ must be identical
because $g_v(\boldsymbol{y})_1 = g_v^{\text{tkn}}(c) = g_v(\boldsymbol{y})_2$. As a result, having $y_1 = y_2$ prevents us from representing any arbitrary $\overline{f}$.

**Response to Reviewer 4.** *Q4-1. Experiment section is not connected to the theoretical part?* We appreciate the
suggestions on experiments by Rev2/Rev4, and we plan to supplement the paper with more experiments on the
assumptions and the BERT architecture in the revision. As for the validity of our assumptions, we note that the poor
performance of the RANDOM pattern partially accounts for the necessity of Assumption 1, because it is very unlikely to
satisfy Assumption 1 with random sparse connections. For a theoretical discussion on the necessity, please see Q2-2.

*Q4-2. Masking and prediction in copying task?* We implemented it as a masked-LM style prediction task by masking
all the tokens in the second half of the sequence. For the test examples, each masked token is predicted independently.
For the reported results we used autoregressive models as in LM. We re-ran experiments with bidirectional models and
observed improved performance (e.g., STAR & 4-layer: 31.19% → 83.57%); we plan to report it in the final version.

*Q4-3. Other questions:* In LM with STAR pattern, tokens cannot attend to the last relay token. We'll add BERT
experiments to make fairer comparisons. In MT, we use the encoder-decoder architecture, and sparsity for both parts.

[Meta-Review · NeurIPS 2020]

The AC and reviewers appreciate the author feedback and thoroughly discussed the paper. The author response brings in important clarifications on how at the approximation is being built and on the terminology (e.g regarding the use of sequences of fixed length for both input and output). We urge the authors to incorporate them in their manuscript.